# Cassini spacecraft reveals global energy imbalance of Saturn

Xinyue Wang[1], Liming Li [2] ✉, Xun Jiang[1], Patrick M. Fry [3], Robert A. West [4], Conor A. Nixon[5], Larry Guan [2], Thishan D. Karandana G [1], Ronald Albright [1], Joshua E. Colwell [6], Tristan Guillot [7], Mark D. Hofstadter [4], Matthew E. Kenyon[4], Anthony Mallama [8], Santiago Perez-Hoyos [9], Agustin Sanchez-Lavega [9], Amy A. Simon [5], Daniel Wenkert [4] & Xi Zhang [10]

The global energy budget is pivotal to understanding planetary evolution and climate behaviors. Assessing the energy budget of giant planets, particularly those with large seasonal cycles, however, remains a challenge without long-term observations. Evolution models of Saturn cannot explain its estimated Bond albedo and internal heat flux, mainly because previous estimates were based on limited observations. Here, we analyze the long-term observations recorded by the Cassini spacecraft and find notably higher Bond albedo ($0.41 \pm 0.02$) and internal heat flux ($2.84 \pm 0.20$ Wm$^{-2}$) values than previous estimates. Furthermore, Saturn's global energy budget is not in a steady state and exhibits significant dynamical imbalances. The global radiant energy deficit at the top of the atmosphere, indicative of the planetary cooling of Saturn, reveals remarkable seasonal fluctuations with a magnitude of $16.0 \pm 4.2\%$. Further analysis of the energy budget of the upper atmosphere including the internal heat suggests seasonal energy imbalances at both global and hemispheric scales, contributing to the development of giant convective storms on Saturn. Similar seasonal variabilities of planetary cooling and energy imbalance exist in other giant planets within and beyond the Solar System, a prospect currently overlooked in existing evolutional and atmospheric models.

The radiant energy budget, which measures the balance or imbalance between the two radiant energy components (i.e., emitted thermal energy and absorbed solar energy), is essential for understanding planets and satellites[1,2]. It significantly affects the thermal structure and related characteristics (i.e., surface properties, atmospheric circulation, weather, and climate) of planets and satellites[3–5]. For giant planets, the radiant energy budget is also used to estimate the internal heat[1,2], which can help us constrain models and theories of planetary formation and evolution[6–10].

A study of Jupiter's energy budget[11] suggests that the wavelength-dependent nature of radiant energy components plays a critical role in determining the energy budget of planets. For Saturn, examination of the radiant energy at the top of the atmosphere is more complicated because of its rings[12–17]. The rings can block and scatter solar radiance

[1]Department of Earth and Atmospheric Sciences, University of Houston, Houston, TX 77004, USA. [2]Department of Physics, University of Houston, Houston, TX 77004, USA. [3]Space Science and Engineering Center, University of Wisconsin-Madison, Madison, WI 53706, USA. [4]Jet Propulsion Laboratory, California Institute of Technology, Pasadena, CA 91109, USA. [5]NASA Goddard Space Flight Center, Greenbelt, MD 20771, USA. [6]Department of Physics, University of Central Florida, Orlando, FL 32816, USA. [7]Université Côte d'Azur, Observatoire de la Côte d'Azur, CNRS, Laboratoire Lagrange, Nice 06108, France. [8]Department of Mathematics and Statistics, University of Maryland, College Park, MD 20742, USA. [9]Departamento de Fisica Aplicada I, Escuela de Ingenieria UPV/EHU, Bilbao 18013, Spain. [10]Department of Earth and Planetary Sciences, UCSC, Santa Cruz, CA 95064, USA. ✉e-mail: lli7@central.uh.edu

and hence affect the amount of solar power absorbed by Saturn's atmosphere. Additionally, thermal emission from Saturn's rings onto its atmosphere influences the radiant energy budget. In addition to directly influencing the radiant energy budget at the top of Saturn's atmosphere, the rings pose difficulties in observing Saturn's atmosphere by blocking the view of an observer to the atmosphere[18].

Moreover, the combination of Saturn's rings and the planet's other unique features, including its large orbital eccentricity (0.052) and axial obliquity (26.7°), results in significant seasonal variations. However, Saturn's seasonal variations of the radiant energy components have not been fully considered in previous studies[12,17,19,20], because long-term high-quality observations were lacking prior to the Cassini mission. The difference between the emitted thermal energy and absorbed solar energy is generally used to estimate the internal heat of giant planets[1,2]. Similarly, the seasonal variations of the radiant energy components have not been fully considered in previous estimates of the internal heat of giant planets including Saturn.

In this study, we use the multi-instrument observations recorded by the Cassini spacecraft during its long-term mission orbiting Saturn (2004-2017) to investigate Saturn's radiant energy budget and its seasonal variations. The investigation of the seasonal variations of Saturn's radiant energy budget is further used to estimate the internal heat.

## Results

### Treatment of rings' effects and computation of radiant energy components

Thermal observations from the Composite Infrared Spectrometer (CIRS)[21] are used to compute Saturn's emitted power, while solar observations from the Imaging Science Subsystem (ISS)[18] and the Visual and Infrared Mapping Spectrometer (VIMS)[22] are used to examine the Bond albedo and hence the absorbed solar power. The Cassini observations offer significant advantages over previous observations in many aspects for measuring the radiant energy budgets of planets and satellites (e.g., better coverage of wavelength, viewing geometry, and observational period)[11,16].

The methodology of using visible and infrared observations to compute the radiant energy components has been described in our

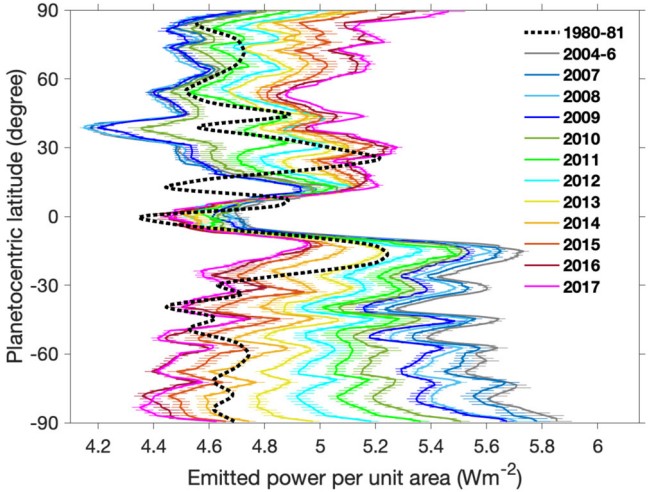

**Fig. 1 | Meridional profiles of Saturn's emitted power during the whole Cassini period (2004–2017).** The profile titled "1980-81" (i.e., the dashed line) comes from previous studies[39–41] based on the observations recorded by Voyager 1 and 2 in 1980-81. The Cassini/CIRS observations in 2004 and 2005 are relatively sparse, so we combine them with the observations in 2006. The thick lines are the profiles of the emitted power in different years. The horizontal error bars represent the uncertainties of measurements, which mainly come from the uncertainty sources from the CIRS data calibration and filling observational gaps.

previous studies[11,16,23–28]. The discussions of the methodology, the corresponding data processing, and the computations of radiant energy budget are presented in the Methods (also see Figs. S1–S25 in the Supplementary Information). Here, we briefly describe the process. We first compute Saturn's emitted power for the Cassini epoch (2004-2017) and extend the results to cover Saturn's complete orbital period from 1995 to 2025, a period including the Cassini epoch. To measure Saturn's bolometric Bond albedo, we need to consider the effects of the rings on the radiant energy components. Based on the model of rings developed by previous studies[12–14] and the optical characteristics of Saturn's rings updated by the Cassini observations[29,30], we calculate the meridional and seasonal distributions of the three effects of the rings (i.e., insolation shadowing, insolation scattering, and thermal emission) on Saturn's atmosphere. The ring-modified solar flux, which is defined as the solar power per unit area, is validated by observations. We use this solar flux as a reference, combined with the solar observations from Cassini and other observatories, to calculate Saturn's Bond albedo. The computed Bond albedo is further used to investigate the global and hemispheric averages of Saturn's absorbed power. It should be mentioned that Enceladus water torus[31,32] can potentially affect the radiant energy budget of Saturn, but its effects are much smaller than those from the rings due to its significantly colder temperature, greater distance from Saturn, and lower density compared to the rings. Therefore, the water torus generated by Enceladus is not considered in our analysis of Saturn's radiant energy budget.

### Emitted power

Figure 1 shows Saturn's emitted power during the Cassini period. The significant increase in emitted power in the middle latitudes of the northern hemisphere (NH) from 2010 to 2011 is related to a giant storm[33–36], which modified the thermal structure of Saturn's atmosphere[35,37,38] and consequently affected the outgoing thermal radiance. In contrast, the southern hemisphere (SH) displays relatively smooth temporal variations, with Saturn's emitted power gradually decreasing from 2004 to 2017. These temporal variations are related to the seasonal transition from summer (2004) to the winter solstice (2017) in the SH. The Cassini profiles shown in Fig. 1 are used to compute the global and hemispheric averages of emitted power for the Cassini epoch (Figs. S3–S5). Figure 1 also displays a comparison of the emitted-power profile between the Cassini measurements and the results from the Voyager spacecraft[39–41]. The Voyager profile differs from all Cassini profiles. Specifically, the solar longitude for the Cassini observations in 2010 is 10.1°, which is close to the solar longitude of 13° for the Voyager observations in 1980–81. This suggests that the two observations are separated by approximately one Saturn year. The difference between the two observations provides an opportunity to examine the possible interannual variations of Saturn's emitted power.

The meridional distribution shown in Fig. 1 is used to compute the global and hemispheric averages of Saturn's emitted power, as presented in Fig. 2. The hemispheric-average emitted powers exhibit much stronger seasonal variations than the global-average quantity because the seasonal variations at the hemispheric scale are partially canceled out when conducting a global average. The 2010 giant storm developed in the middle latitudes of the NH did not affect the thermal structure of the Southern Hemisphere (SH), so the SH-average emitted power displays a monotonic decrease in the Cassini epoch, which corresponds to the decreasing solar irradiance in the SH from 2004 to 2017. The effects of the 2010 giant storm on the emitted power in the middle latitudes of the NH (Fig. 1) is clearly shown in the global and NH averages (Fig. 2).

The Cassini epoch does not cover Saturn's complete seasonal cycle, so we extrapolate the measurements from the Cassini epoch (2004-2017) to a complete orbital period (1995-2025), including the Cassini epoch. Solar irradiance, which has a clear seasonal cycle, is the most important factor affecting the thermal property and, hence, the

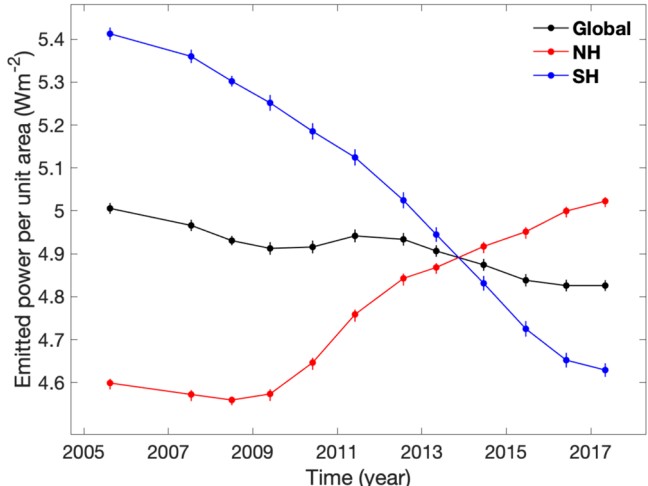

**Fig. 2 | Global and hemispheric averages of Saturn's emitted power during the Cassini epoch.** Vertical lines represent the measurement uncertainties. The NH and SH represent the northern and southern hemispheres, respectively.

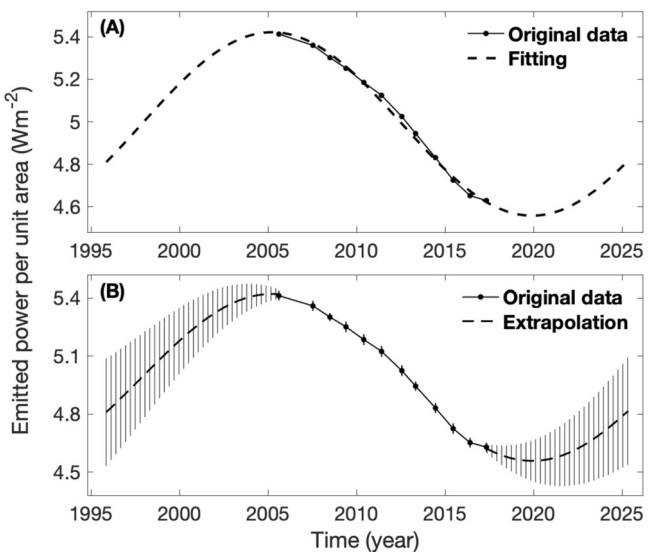

**Fig. 3 | Fitting Saturn's SH-average emitted power and extrapolating it from the Cassini epoch to the complete orbital period. A** The observed emitted power is fitted by a sine function with a fixed period of one Saturn year (29.4 years). **B** The fitting in panel **A** is used to extrapolate the emptied power from the Cassini epoch (2004-2017) to the complete orbital period (1995-2025), including the Cassini epoch. Vertical lines in panel **B** represent uncertainties.

emitted power. Additionally, Earth's emitted power[42,43] displays a clear seasonal cycle. Here, we assume that Saturn's emitted power also has a seasonal cycle. We use a sine function with a period of 29.4 years (i.e., Saturn's orbital period around the Sun) to fit the emitted power. The test of the SH-average emitted power (Fig. 3), which was not affected by the 2010 giant storm, suggests that such a fitting does a good job of fitting Saturn's emitted power. The fitting shown in Fig. 3 is used to extend Saturn's emitted power from the Cassini epoch (2004-2017) to the entire orbital period (1995-2025).

Estimating the uncertainty of such extrapolation is difficult. We combine the uncertainty in the measurements of the emitted power based on the Cassini observations (Fig. 2) with the sine-function fitting residuals (i.e., fitting results minus observational data) to estimate the uncertainties of the extrapolated emitted power for the years neighboring to the Cassini epoch. For example, the uncertainty of the 2017

measurement of Saturn's emitted power is combined with the sine-function fitting residual for the 2017 observational data to estimate the uncertainty of the extrapolated emitted power in 2018. Likewise, we can combine the measurement uncertainty and the sine-function fitting residual at the 2005 observational data to estimate the uncertainty of the extrapolated emitted power in 2004.

The above estimates probably work for these extrapolated emitted powers at times neighboring the Cassini epoch (2004 and 2018, respectively), but they may underestimate the uncertainties for the extrapolated emitted powers with times far away from the Cassini epoch. Here, we use the standard deviation of the measured emitted powers during the Cassini epoch, which is much larger than the above-mentioned uncertainties, as the upper limit of the uncertainties for the extrapolated emitted power. Given their distance from the Cassini epoch, the initial and final years of the entire orbital period (1995 and 2025, respectively) are expected to bear the greatest uncertainty, corresponding to the standard deviation during the Cassini epoch. Subsequently, we linearly interpolate the uncertainty of emitted power at the beginning of the Cassini epoch (2005) and the largest uncertainty in the initial year of the whole orbital period (1995) to get the uncertainties for the extrapolated emptied powers from 1995 to 2004. Likewise, we linearly interpolate the uncertainty of emitted power at the end of the Cassini epoch (2017) and the largest uncertainties at the final year of the whole orbital period (2025) to get the uncertainties for the extrapolated emptied powers from 2018 to 2025 (see Fig. 3).

Fitting the global and NH averages of the emitted power is more challenging because the 2010 giant storm affected both the NH and global emitted powers. A study[25] suggests that the 2010 giant storm increased the global-average emitted power by 2%. We first subtract such an increase (2%) from the observed global-average emitted power to get a modified emitted power for the period after the 2010 giant storm (i.e., 2011-2017), as shown in panel A of Fig. 4. We then fit the modified emitted power using a sine function with a period of 29.4 years. Panel B of Fig. 4 shows that this fitting works well. Finally, we use the fitting results to extrapolate the global-average emitted power from the Cassini epoch to the entire orbital period (panel C of Fig. 4). For the period after 210, we increase the modified emitted powers and the corresponding fitting results by 2% to revert the modified emitted powers from 2010 to 2017 back to the original measurements. Similarly, we modify the NH-average emitted power and use the fitting function to extrapolate the NH-average emitted power from the Cassini epoch to the complete orbital period (see panels D-F of Fig. 4). Extrapolating to the complete orbital period can help us better understand the seasonal variations of Saturn's emitted power. Additionally, it can help us refine the internal heat flux.

Considering the large uncertainties in extrapolating the analysis from the Cassini epoch to the whole orbital period (see Figs. 3 and 4), we try to validate the extrapolation. A previous investigation of Saturn's global-average emitted power[19] is based on relatively high-quality observations conducted in 1971-72. Unfortunately, the observational times are beyond the whole orbital period including the Cassini epoch (1995–2025). We add the observational time (1971-72) by a one orbital period (29.4 years) to project the study to year 2001 and then compare with our extrapolation of the global-average emitted power (the green point in panel C in Fig. 4). The comparison shows that the difference between the previous result[19] and the extrapolated value is less than the uncertainty of the extrapolated value, which suggests that they are qualitatively consistent.

We also examine observations and studies of Saturn's thermal radiance after the Cassini epoch. A study[44], based on the ground-based mid-infrared observations, analyzed the brightness temperature of Saturn. Specifically, this study provides the brightness temperature recorded at 17.7 μm from 2017 to 2022, which is a period following the Cassini epoch. Additionally, the 17.7 μm observations sound the upper troposphere. Our previous study[16] suggests that Saturn's emitted

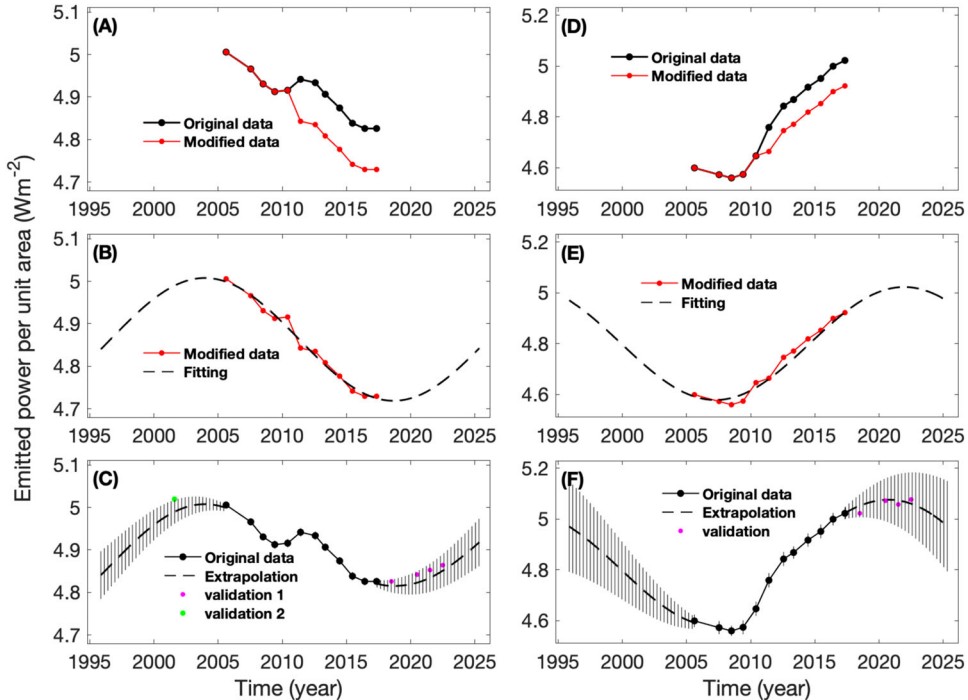

**Fig. 4 | Fitting Saturn's global-average and NH-average emitted powers and extrapolating them from the Cassini epoch to the entire orbital period.** **A** Modified global-average emitted power. The original emitted power was modified by removing the effect of the 2010 giant storm. **B** Fitting the modified emitted power by a sine function with a fixed period of one Saturn year (29.4 years). **C** Extrapolating the global emitted power from the Cassini epoch (2004–2017) to the entire orbital period (1995-2025). The fitting results shown in panel **B** are used

for the extrapolation. The results in (**D**–**F**) are the same as the global analyzes in (**A**–**C**) respectively except for the NH. Vertical lines in panels **C** and **F** represent uncertainties. The global-average emitted power before the Cassini epoch (i.e., "validation 2" with the green dot in panel **C** comes from a previous study[19]. The emitted powers after the Cassini epoch (i.e., "validation 1" with magenta dots in panel **C** and "validation" with magenta dots in panel **F**) are based on the ground-based observations of Saturn's brightness temperature[44].

power mainly comes from the upper troposphere, so the 17.7 μm observations can be used to estimate Saturn's emitted power.

The post-Cassini period from 2017 to 2022 corresponds to the summer season of the NH. The view geometry of the ground-based observatories, along with the effects of rings (e.g., blocking), make it challenging to observe the entire SH. Consequently, the ground-based observations[44] from 2017 to 2022 cover latitudes from 10°S to 90°N, which do not resolve the whole SH. We first computed the NH-average brightness temperature to validate the extrapolation of NH-average emitted power. Subsequently, we used the average over limited latitudes in the SH (0-10°S) to represent the SH-average brightness temperature. This was then combined with the NH-average brightness temperature to estimate the global-average emitted power.

While brightness temperature is related to effective temperature (i.e., emitted power), they are not equal[16]. Our focus here is on examining the temporal variations of our extrapolation of emitted power, so we utilize the temporal variations of brightness temperature to estimate the temporal variations of effective temperature (i.e., emitted power). We use NH and global averages of brightness temperature from 2017 to 2022 to represent effective temperature and then compute the corresponding emitted power. Since brightness temperature is not equal to effective temperature[16], the computed emitted powers from 2017 to 2022 are scaled to the finally estimated emitted power using the ratio between the emitted power computed from brightness temperature and the Cassini-CIRS measured emitted power in 2017. In other words, we shift the computed emitted powers based on ground-based observed brightness temperature (2017-2022) to align the computed 2017 emitted power with the corresponding CIRS-measured emitted power. Subsequently, the finally estimated emitted power at the global and NH scales are compared to the extrapolations, as shown in panels C and F of Fig. 4. The temporal variations of the estimated

emitted power from brightness temperature are generally consistent with our extrapolated emitted power based on the Cassini/CIRS measurements. This suggests that observations after the Cassini epoch qualitatively validate the extrapolation of Saturn's emitted power.

## Bolometric bond albedo

The absorbed power is controlled by the Bond albedo, which is further determined by the full-disk reflectance (i.e., the ratio between the reflected solar radiance and the incoming solar radiance of the full disk for a planet)[26]. As discussed in the Methods (see subsection "Observations of Saturn's Full-disk Reflectance"), the measurements of Saturn's Bond albedo are mainly based on the observations recorded by the two instruments onboard the Cassini spacecraft (i.e., ISS and VIMS). Among the two instruments, the ISS observations are used to investigate the phase function of Saturn's full-disk reflectance (i.e., the function of Saturn's full-disk reflectance varying with phase angle) because they have the best coverage of phase angle among all available observations. However, the coverage of phase angle for the ISS observations in each year of the Cassini epoch (2004-2017) is not enough to measure phase function and hence Bond albedo (Fig. S12). Additionally, our investigations of Saturn's full-disk at different times (Figs. S13 and S14) suggest that Saturn's full-disk reflectance did not significantly vary with time during the Cassini epoch. Therefore, we combine all the ISS observations from the Cassini period to explore the time-mean full-disk reflectance and, hence, the phase function. Figure 5 shows the ISS measurements of Saturn's full-disk reflectance at these ISS filters (wavelengths) with enough coverage of phase angle for investigating the phase function.

The ground-based full-disk observations recorded by the European Southern Observatory (ESO)[45] (Fig. S17) have a phase angle of 5.7°, which can help fill observational gaps in low phase angles for the

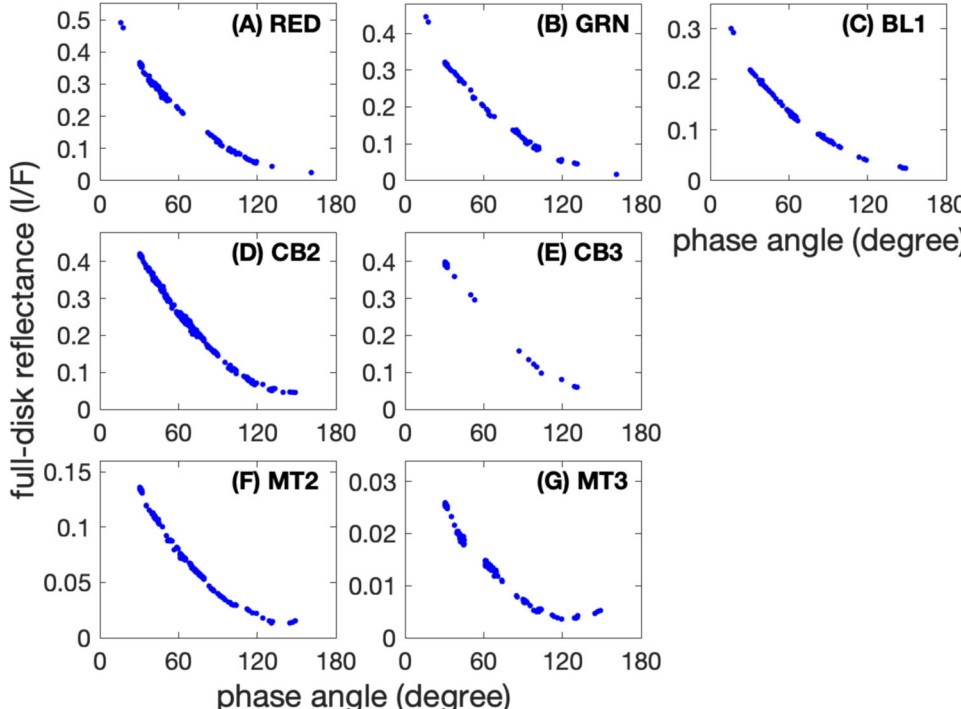

**Fig. 5 | Saturn's full-disk reflectance at different phase angles recorded by the Cassini/ISS 7 filters. A** RED filter (647 nm); **B** GRN filter (568 nm); **C** BL1 filter (463 nm); **D** CB2 filter (752 nm); **E** CB3 fitler (939 nm); **F** MT2 filter (728 nm); and **G** MT3 fitler (890 nm).

Cassini/ISS measurements. Therefore, we combine the measurements of the full-disk reflectance based on the ESO observations with the Cassini/ISS measurements to investigate the phase functions of Saturn's full-disk reflectance. However, the combined observations still have gaps in phase angle and wavelength that need to be filled before computing the Bond albedo. Our study of Jupiter's Bond albedo[11] suggests that a polynomial function with the least-squares technique[46] works well for fitting the phase function of the full-disk reflectance for the gas giants, which can be used to fill the observational gaps in phase angle. We try different polynomial functions to fit the combined measurements of Saturn's full-disk reflectance and find that a four-order polynomial function $P(g) = c_1 g^4 + c_2 g^3 + c_3 g^2 + c_4 g + c_5$ (where P and g represent phase function and phase angle, respectively, and the parameters $c_1$, $c_2$, $c_3$, $c_4$, and $c_5$ are fitting coefficients to match the observations with the least-squares technique) has the smallest fitting residual. It should be mentioned that we also tested some physically-based functions, such as the double Henyey-Greenstein (H-G) function[47,48]. Figure 6 shows the comparison of fitting between our polynomial function and the double H-G function, which suggests that the fittings of polynomial function have relatively small fitting residuals (i.e., fitting results minus observational data). The physically-based functions can shed light on the atmospheric properties of giant planets[49], but here we focus on filling the observational gaps in phase angle and hence computing Saturn's Bond albedo. Therefore, the polynomial-function fitting is used in our analysis. The fitting residuals, which are combined with the calibration errors, are utilized to estimate the uncertainties of phase function and hence Bond albedo (see Methods, subsection "Measurements of Saturn's Bond Albedo").

Figure 7 displays the fitting results for the data of combining the ESO and Cassini/ISS measurements at ISS seven filters /wavelengths (i.e., GRN at 568 nm, RED at 647 nm, BL1 at 463 nm, CB2 at 752 nm, CB3 at 939 nm, MT2 at 728 nm, and MT3 at 890 nm). The fitting results are used to fill the observational gaps in phase angle. It should be mentioned that both the data and the fitting show that Saturn's full-disk reflectance increases with phase angle when phase angles become very large (i.e., >130°) at some wavelengths (e.g., panels F and G for the two

methane-absorption filters), which is probably related to the moderately forward scattering of the haze particles in Saturn's atmosphere[50]. The fitting at these high phase angles (150-180°), where observations are lacking, may have relatively large uncertainty. But the reflectance at these phase angles does not significantly contribute to the Bond albedo because there is a factor of sine of phase angle in computing Bond albedo[1,2,11,26] and this factor is small when phase angles approach to 180°. Therefore, the fitting uncertainties at the high phase angles, which are considered in our estimate of total uncertainty (see Methods, subsection "Measurements of Saturn's Bond Albedo"), do not significantly affect our analysis of Saturn's Bond albedo.

The fitting presented in Fig. 7 generates complete phase functions at the wavelengths recorded by the ISS seven filters. Here, we discuss how to interpolate/extrapolate the complete phase functions from the ISS seven wavelengths to the entire spectral range under investigation. As shown in Fig. S17, the observations from IUE[51], Aerobee[52], and ESO[45] provide the spectra of full-disk reflectance in the wavelengths 195-244 nm, 245-305 nm, and 300–1050 nm, respectively. Additionally, the Cassini/VIMS recorded the reflectance spectra from 350 nm to 5130 nm (Fig. S16). By combining the complete phase functions at the ISS seven wavelengths with the available reflectance spectra in other wavelengths, we can fill in the observational gaps in other wavelengths by interpolation and extrapolation (see Methods, subsection "Measurements of Saturn's Bond albedo").

Figure 8 displays the full-disk reflectance in the domain of wavelength and phase angle. By integrating the reflectance over phase angle and averaging it over wavelength, we find that Saturn's Bond albedo is 0.41±0.02. The most thorough previous analysis[12] suggested a value of 0.34±0.03 for Saturn's Bond albedo. The difference of Bond albedo between the Cassini epoch (2004-2017) and Voyager time (1980-1981) should be investigated. First, we examine the possibility of temporal variation of Saturn's Bond albedo from the Voyager time to the Cassini epoch. The Cassini analysis (Figs. S13 and S14) reveals that Saturn's full-disk reflectance and Bond albedo did not significantly vary during the Cassini epoch. Additionally, Earth-based observations[45,53] also suggest that the Saturn's full-disk reflectance and, hence, Bond albedo basically

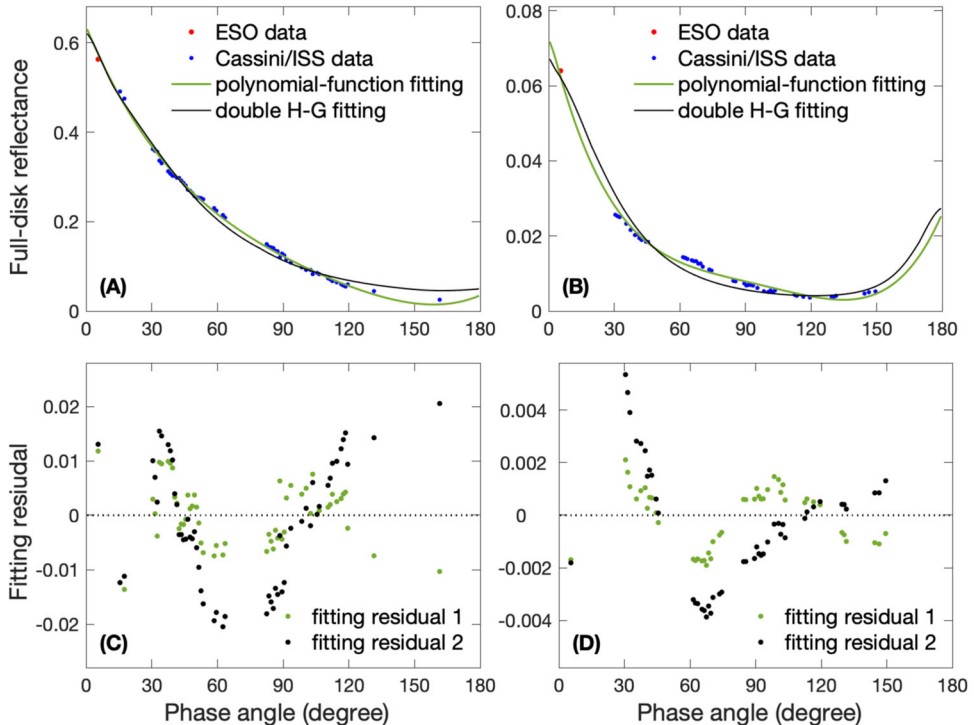

**Fig. 6 | Examples of fitting the phase function of Saturn's full-disk reflectance.** **A** Fitting results for the combined data between the ESO observations and the Cassini ISS observations recorded at the RED filter (647 nm). A four-order polynomial function (green line) and the double H-G function (black line) are used for fitting the data. **C** Fitting residuals (i.e., fitting results minus observational data) for the fittings shown in (**A**). Fitting residual 1 is for the fitting with a four-order polynomial function and fitting residual 2 is for the fitting with the double H-G function. Panels **B** and **D** are the same as panels **A** and **C** respectively except for the combined data between the ESO observations and the Cassini ISS observations recorded at the strongest methane-absorption filter (i.e., MT3 at 890 nm).

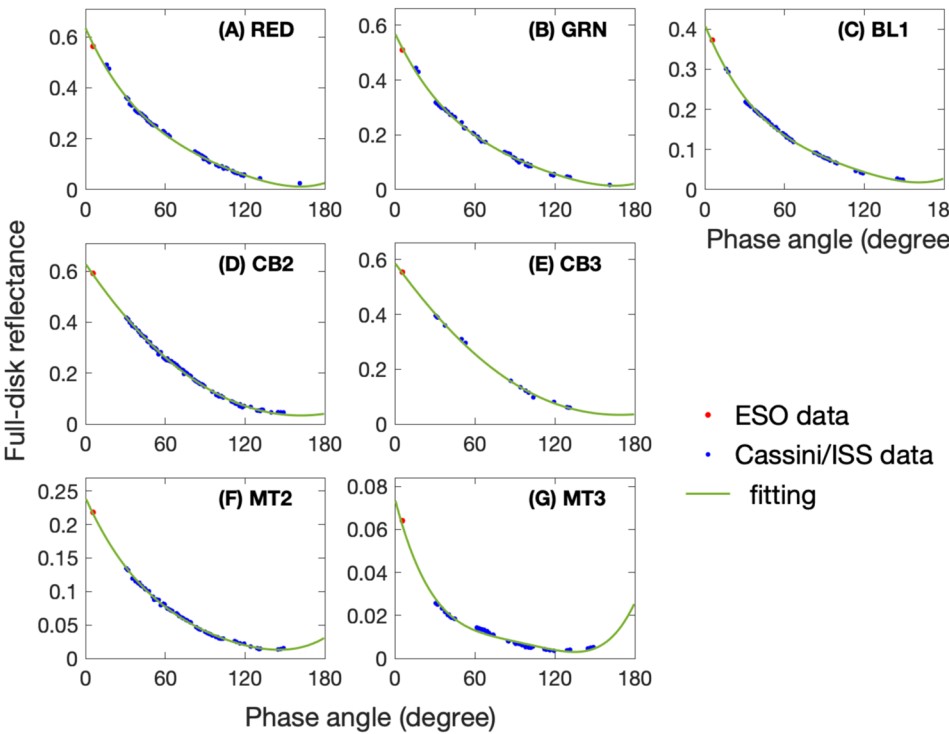

**Fig. 7 | Fitting results for the combined measurements obtained using the ESO observations and the Cassini/ISS observations recorded at the seven filters.** **A** RED filter (647 nm); **B** GRN filter (568 nm); **C** BL1 filter (463 nm); **D** CB2 filter (752 nm); **E** CB3 filter (939 nm); **F** MT2 filter (728 nm); and **G** MT3 filter (890 nm).

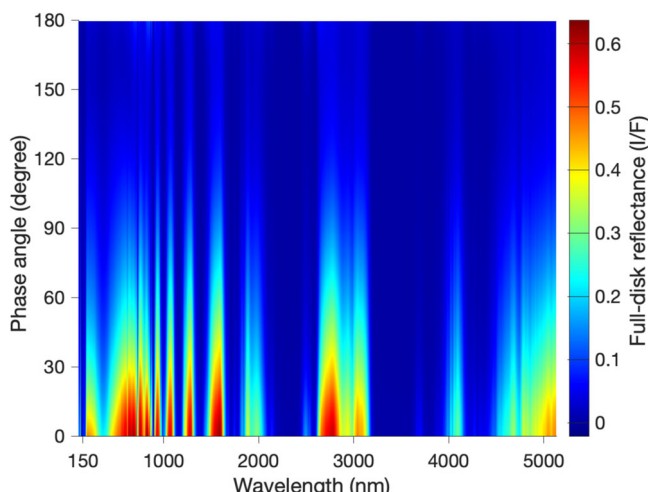

**Fig. 8 | Full-disk reflectance of Saturn in the two-dimensional domain of phase angle and wavelength.** The phase angle is defined as the angle between the line from the Sun to Saturn and the line from Saturn to Cassini. It varies in a complete range of 0–180°. Wavelength changes from 120 nm to 5131 nm, where the data are available. The solar spectral irradiance at wavelengths from 120 nm to 5131 nm occupies approximately 99.5% of the total solar flux. Results are mainly based on the observations recorded by the Cassini spacecraft (ISS and VIMS) and ESO.

remained constant from 1993 to 1995. Finally, a study[54] also indicates that there was no significant difference in Saturn's full-disk reflectance between the observations during the 1963-1965 period[55,56] and those during the 1991-2009 period[57]. These studies suggest that Saturn's full-disk Bond albedo did not significantly change from the Voyager epoch (1980-1981) to the Cassini time (2004–2017). Therefore, we think that the difference in Bond albedo between the Voyager and Cassini measurements is caused by something else.

The Voyager analysis[12] was based on the observations covering a latitude band from −11° to −32° in the SH. Such a latitude band sits in a belt (Fig. S22). Belts, which correspond to areas lacking clouds, have lower reflectance than that of zones, which are full of highly reflective clouds[41]. As a result, the Voyager study may have underestimated Saturn's Bond albedo. Our analysis suggests that the reflectance decreased by 4.6% from the global-average reflectivity to the reflectance of the belt covering latitudes from −11° to −32°. The difference of Bond albedo between the Cassini measurements (0.41 +/−0.02) and the Voyage analysis (0.34 +/−0.03) has a lower limit (0.37−0.39)/0.37 = −5.4% and an upper limit (0.31−0.43)/0.31 = 38.7%. The difference of reflectance between the global average and the belt from −11° to −32° is close to the lower limit of the Bond-albedo difference between the Cassini and Voyager measurements, but it cannot explain all the difference. Reproducing Voyager's analysis of Saturn's Bond albedo and further discerning the differences between Voyager's analysis and the Cassini measurements is challenging due to the lack of necessary details in analyzing the Voyager data in the previous study[12] and the difficulty of reprocessing the old Voyager datasets. It should be mentioned that our study is based on long-term multi-instrument Cassini observations, which exhibit significant improvements in various aspects (such as much better coverage of latitude, wavelength, and phase angle) compared to the Voyager observations used by the previous study[12]. Therefore, we think that the Bond albedo generated by our study is more robust.

### Radiant energy budget and internal heat

The Cassini analysis (Figs. S13 and S14) and previous investigations[45,53] suggest Saturn's full-disk reflectance did not significantly vary with time. Therefore, we assume that Saturn's Bond albedo is constant throughout the entire orbital period including the Cassini epoch

(1995–2025). Then we can combine the Bond albedo (0.41±0.02) with the rings-modified global-average solar flux (panel A of Fig. 9) to compute the absorbed power (panel B of Fig. 9), which is compared to the global-average emitted power (panel B of Fig. 9). The difference between the global-average absorbed and emitted powers, equivalent to the energy budget of Saturn, is presented in panel D of Fig. 9. The absorbed solar power, ranging from 1.80 to 2.37 Wm⁻², is much smaller than the emitted power, varying from 4.83 to 5.01 Wm⁻². Therefore, there is a significant radiant energy deficit, which is defined as the difference between the absorbed solar energy and the emitted thermal energy (i.e., the absorbed solar energy minus the emitted thermal energy), for Saturn as a whole (including its atmosphere and interior)[12,17,19,20]. This energy deficit suggests that Saturn is losing energy, a phenomenon referred to as global cooling. The range of the energy deficit spans from −2.63 ± 0.08 Wm⁻² in 2003 to −3.05 ± 0.07 Wm⁻² in 2013. The seasonal variations of the energy deficit imply that Saturn's radiant energy budget and global cooling are not stable over time. Saturn's global energy budget is assumed to maintain equilibrium across all time scales in current models and theories[10,58–60] and the new findings can help to better develop these models and theories.

Three principal factors drive the time variability of the radiant energy budget. The first aspect pertains to Saturn's large eccentricity (0.052), which results in significant variations in the global-average solar constant, defined as the solar flux without accounting for the effects of the rings (indicated by the red line in panel A of Fig. 9). The solar constant at Saturn changes by approximately 24.3% from aphelion to perihelion. Secondly, the rings create a large modulation in the seasonal variations of the global-average solar flux with a magnitude of 10.7% (panel A of Fig. 9), which consequently impact the absorbed solar power (panel B of Fig. 9). Lastly, the occurrence of Saturn's giant convective storms, approximately every thirty years[61,62], modify the emitted power and the absorbed solar power of Saturn. A previous study[25] and Fig. S14 suggest that the 2010 giant storm changed the global-average emitted power and absorbed power by 2.0% and 2.9%, respectively. Other potential factors, such as small and mesoscale storms and waves, which may have relatively minor effects on the temporal variations of the radiant energy budget, are not considered in this study. At the global scale, the seasonal variations are considerably more pronounced in the absorbed power (panel B of Fig. 9) than in the emitted power (panel C of Fig. 9). As a result, the difference between the two powers (panel D of Fig. 9) largely mirrors the seasonal fluctuations observed in the absorbed power (panel B of Fig. 9).

The difference between the absorbed and emitted powers can also be used to estimate the internal heat. The interior evolution of giant planets has much longer time scales than their orbital periods[63]. Therefore, Saturn's internal heat has been assumed to be constant throughout its orbital period[12,17,19,20]. On the other hand, our analysis suggests that the radiant energy budget significantly varies during an orbital period (Fig. 9). This variation must be considered when estimating the seasonally constant internal heat. Such seasonal variations were not fully explored in previous studies of giant planets[12,17,19,20], leading to less precise estimates of their internal heat. Therefore, the internal heat of giant planets should be re-examined.

The long-term Cassini observations partially covered three of Saturn's seasons (i.e., part of the NH winter, complete NH spring, and part of NH summer), providing a great opportunity to estimate Saturn's internal heat. Based on Fig. 9, we first calculate the time-average radiant energy components over the entire orbital period (1995–2025): 2.04 ± 0.17 Wm⁻² and 4.88 ± 0.11 Wm⁻² for the absorbed and emitted powers, respectively. The difference between the time-average absorbed and emitted powers is used to estimate the internal heat flux, yielding a value of 2.84 ± 0.20 Wm⁻² for the internal heat flux. We also examine the internal heat flux based on the global-average energy during the Cassini epoch (2004–2017) only, which yields a value of 2.89 ± 0.18 Wm⁻². The period between the perihelion (July, 2003) and

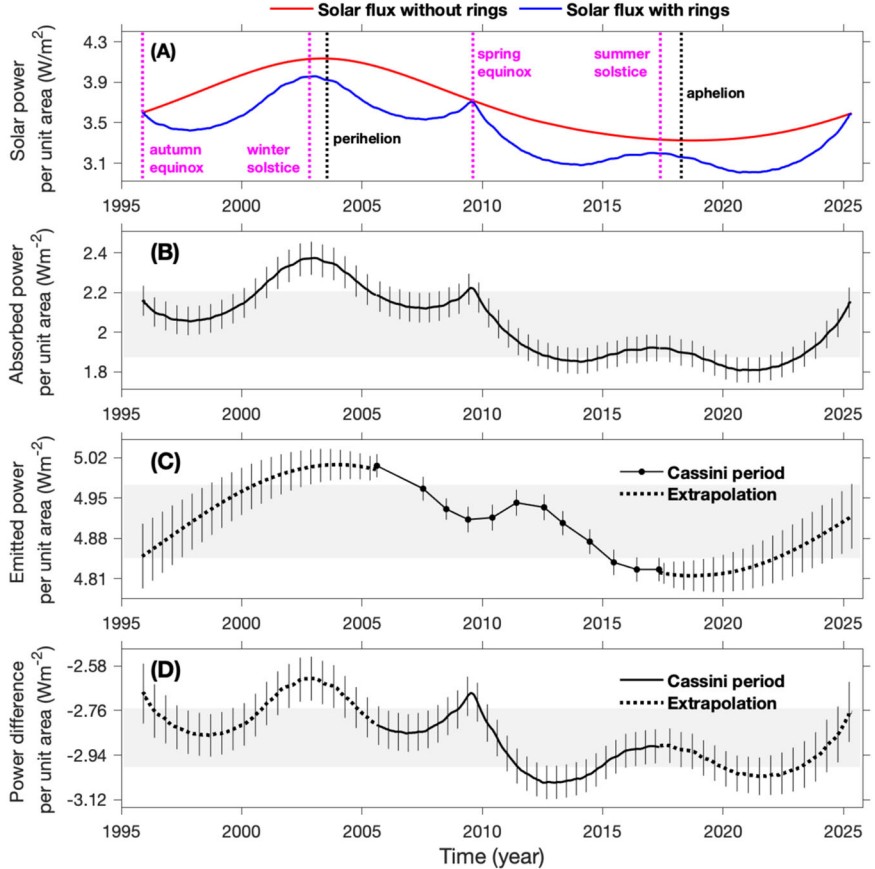

**Fig. 9 | Saturn's global-average solar power, absorbed power, emitted power, and the difference between absorbed and emitted powers. A** The comparison of solar power per unit area (i.e., solar flux) between two scenarios: (1) computation without considering the two effects of the rings (i.e., insolation shadowing and scattering); and (2) calculation including the effects of the rings (Fig. S6). The four magenta vertical dashed lines (from left to right) indicate the autumn equinox, winter solstice, spring equinox, and summer solstice of the NH, respectively. The two black vertical dashed lines represent Saturn's perihelion and aphelion in its orbit around the Sun. **B** The absorbed power. The absorbed power is a combination of the absorbed solar power (Fig. S25) and the atmosphere's absorption of the

thermal radiation from the rings (Fig. S5), although the latter is considerably smaller in magnitude compared to the former. **C** The emitted power. The emitted power comes from panel **C** of Fig. 4. **D** The difference between the absorbed and emitted powers (i.e., the absorbed power minus the emitted power). The solid lines represent the measurements taken during the Cassini epoch (2004–2017), while the dashed lines display the extrapolated results for the entire orbital period (1995-2025). The gray rectangular areas in panels (**B**–**D**) illustrate the variances of the qualities depicted in these panels. Specifically, the top boundary of each rectangle area represents the time-mean value plus the standard deviation, while the bottom boundary represents the time-mean value minus the standard deviation.

the aphelion (April, 2018) is roughly consistent with the Cassini epoch, suggesting that the Cassini observations have captured the dominant variations in the seasonal cycle of solar constant at Saturn. The seasonal cycle of solar constant is one of the key factors affecting the seasonal variations of Saturn's radiant energy budget[15,16]. It helps explain the consistency of the estimated time-average internal heat between the complete orbital period (2.84 ± 0.20 Wm⁻²) and the Cassini epoch (2.89 ± 0.18 Wm⁻²). The new value of internal heat (2.84 ± 0.20 Wm⁻²), accounting for the seasonal variations in Saturn's radiant energy budget, is notably higher than the previous best estimate (2.01 ± 0.14 Wm⁻²) derived from the limited Voyager observations[12]. The new measurements of internal heat can help us to constrain the theories of planetary formation and evolution. The evolution models that used the previously estimated albedo of Saturn (0.34±0.03), alongside phase diagrams consistent with Jupiter/Galileo helium constraint, overpredicted the previously estimated internal heat flux (2.01 ± 0.14 Wm⁻²) at the solar age[10]. It was suggested that the model can explain the evolution of Saturn better if the Bond albedo value is higher, such as 0.5, indicating a larger internal heat. These values are more compatible with the results of this study (Bond albedo 0.41 ± 0.02 and internal heat flux 2.84 ± 0.20 Wm⁻²) than the estimates (Bond albedo 0.34 ± 0.03 and internal heat flux 2.01 ± 0.14 Wm⁻² from the Voyager study[12].

## Energy imbalance of upper atmosphere

For the system of Saturn including both atmosphere and interior, the absorbed and emitted energies are the only energy source and sink (Fig. 9). But for Saturn's upper atmosphere including the weather layer, the internal heat also serves as an energy source[64]. As a result, we can merge the seasonally-constant internal heat flux with the seasonally-fluctuating absorbed solar power to calculate the overall input energy flux for Saturn's upper atmosphere. Conversely, the seasonally-varying emitted power constitutes the sole output power from the upper atmosphere. The comparison between the input and output fluxes can be used to examine the energy budget of Saturn's upper atmosphere. Panel A of Fig. 10 suggests a significant energy imbalance at the global scale during the Cassini epoch. Specifically, the global energy imbalance changed from an excess of 4.0 ± 1.6% of the emitted power in 2009 to a deficit of −3.4 ± 1.4% of the emitted power in 2013. For the entire orbital period, the global energy imbalance can be even larger (e.g., an energy excess 5.0 ± 1.8% of the emitted power in 2003).

Saturn's atmosphere does not exhibit significant differences in reflectance between the two hemispheres (Figs. S23 and S24), so we assume that the two hemispheres have the same Bond albedo. Additionally, we assume that the internal heat does not significantly vary from the NH to the SH. Therefore, we can explore the energy budget at the hemispheric scale, as shown in panels B and C of Fig. 10. The

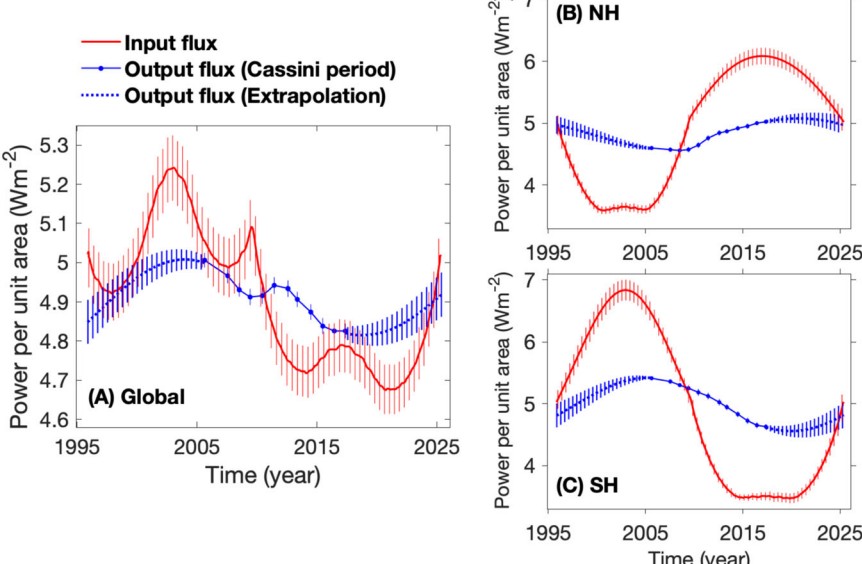

**Fig. 10 | Saturn's energy budgets of the upper atmosphere at both global and hemispheric scales.** The energy budget is determined by comparing the input flux to the output flux. The vertical lines represent uncertainties in the measurements of the two fluxes. The input flux is calculated by combining the absorbed power in the entire orbital period (panel **B** of Fig. 9) with the internal heat flux calculated as described in the text. The output flux is determined solely by the emitted power. The global and hemispheric averages of emitted power come from Figs. 3 and 4. **A** The comparison between the global-average input and output fluxes. **B** and **C** are the same as (**A**) except for the hemispheric analyzes of the NH and SH respectively.

hemispheric-average energy budget displays much larger energy imbalances than that of the global-average energy budget. The NH-average energy imbalance can vary from a deficit $-25.2 \pm 1.5\%$ of the emitted power in 2000 to an excess $21.8 \pm 2.1\%$ of the emitted power in 2015. For the SH, the hemispheric-average energy imbalance ranges from an excess $27.8 \pm 2.6\%$ in 2002 to a deficit $27.9 \pm 1.3\%$ of the emitted power in 2014. Panel B further suggests that the NH has an energy excess in its spring and summer seasons (2009–2025) and an energy deficit in its autumn and winter seasons (1995–2009). The NH and SH have opposite seasons, so panel C suggests that the SH also has an energy excess in its spring and summer seasons and an energy deficit in its autumn and winter seasons. The energy excess mainly occurs in the spring and summer seasons of the two hemispheres, which can be explained by the fact that the two seasons have a relatively large solar flux in the two hemispheres (Fig. S7).

The energy imbalances revealed in Fig. 10 may help us better understand planetary atmospheres. Hemispheric energy imbalances, with an excess of over 20% of the emitted power, appear in the spring and summer seasons of both hemispheres. This excess energy probably contributes to the development of Saturn's storms, especially the periodic giant storms[61,62]. The historical survey of Saturn's storms[61] reveals that Saturn's storms, regardless of their size, basically occur in the spring and summer seasons of the two hemispheres. This spatio-temporal distribution of storms is consistent with the seasons and hemispheres with energy excess (Fig. 10), which suggests that energy excess is probably a necessary condition for storms on Saturn. Saturn's giant storms mainly occur during the spring and summer seasons of the NH[61,65], which corresponds to the NH energy excess at the same time. The NH spring and summer seasons (e.g., 2009-2025) are around the time when Saturn moved through the aphelion in its orbit around the Sun (e.g., 2018), so Saturn's giant storms mainly develop in the aphelion seasons. There is ample evidence for a relationship between radiant energy imbalance and storms on other planets. For Mars, relatively large dust storms mainly occur in the perihelion seasons with maximal global energy excess[66]. Large storms appear to develop in different seasons between Saturn (aphelion seasons) and Mars (perihelion seasons). This difference may be because Saturn has a thick

atmosphere without a solid boundary, while Mars has a very thin atmosphere (surface pressure approximately 0.01 bar) with a solid surface. Mars' radiant energy excess can directly drive storms in its thin atmosphere by heating its solid surface. On Saturn, the giant storms are probably rooted in the relatively deep atmosphere at the level of water clouds around 5–10 bars, while the radiant energy imbalance mainly happens around the top ammonia cloud level around 0.5 bar. Linking the radiant energy imbalance at the top cloud level to deep moist convection, which is thought to be a driving mechanism for giant storms on Saturn, needs further theoretical and model studies.

## Discussion

The temporal variations of energy imbalance revealed in this study should be taken into consideration when developing the models and theories of Saturn's atmosphere. In addition, it has been shown that the spatial and temporal inhomogeneities of the atmospheres on giant planets, such as those induced by orbital eccentricity, obliquity of the rotation axis, and rings' effects, would boost the planetary cooling flux and energy imbalance compared with the traditionally assumed one-dimensional evolution models[67,68]. Re-examining the cooling fluxes, internal heat, and energy imbalances of other giant planets in our solar system by considering the seasonal variation of radiant energy components is crucial to understand planetary formation and evolution, both within and beyond our solar system. Jupiter and Uranus have orbital eccentricities of 0.049 and 0.047, respectively, which are comparable to Saturn's orbital eccentricity (0.052). As a result, the seasonal variations in solar irradiance for the three giant planets (Jupiter, Saturn, and Uranus) have a similar magnitude (approximately 20%). It is anticipated that Jupiter and Uranus also exhibit considerable seasonal energy imbalances at both global and hemispheric scales. Neptune has a much smaller orbital eccentricity (0.010), and therefore, its global-average seasonal variations in solar irradiance are relatively small (approximately 4%). However, Neptune has a significant obliquity (28.3°), implying that it experiences a hemispheric-average seasonal energy imbalance caused by its obliquity.

Especially, Uranus is a unique planet in our solar system due to its large orbital eccentricity (0.047) and very high obliquity (97.8°), which

likely result in the strongest seasonal energy imbalance. Addressing this energy imbalance can help reconcile a long-term discrepancy in estimates of Uranus' internal heat between observational investigations[69,70] and theoretical studies[71]. A flagship mission to Uranus, which was given the highest priority in the recent decadal survey of planetary science and astrobiology[72], will provide an opportunity to investigate the predicted significant energy imbalance and re-examine the internal heat of this planet. Moreover, many extrasolar giant planets like HD 80606 b display high eccentricities, and there is a possibility that some of them might harbor ring systems[73]. These planets could also undergo substantial fluctuations in their global energy budgets. It is imperative to take this significant factor into careful consideration when developing their evolutionary and climate models - a suggestion inspired by our current findings of Saturn.

## Methods

### Methodology for computing radiant energy budget

The radiant energy budget of a planet is determined by the amount of thermal energy emitted by the planet and the solar energy it absorbs[1,2,11,16,26]. The methodology for computing Saturn's emitted power using Cassini CIRS data was presented in one of our previous studies[16]. The basic idea is to integrate the recorded thermal radiance over different emission angles to determine the emitted power[1,16]. The absorbed solar energy is determined by the full-disk Bond albedo with the known solar irradiance at the distance of Saturn[1,2]. The methodology of computing the full-disk Bond albedo using Cassini observations is presented in our previous studies of the Bond albedos of Jupiter, Titan, and Enceladus[11,26,28]. However, the rings of Saturn introduce difficulties in measuring the Bond albedo. Therefore, the effects of the rings on the radiant energy budget must be addressed first. After that, the observations recorded by the Cassini ISS and VIMS are used to measure Saturn's Bond albedo and, consequently, the absorbed solar power. The details of computing the radiant energy budget with the above-mentioned methodology and the Cassini observations are described below.

### Measurements of Saturn's emitted power

The measurements of Saturn's emitted power are based on the infrared spectra recorded by the CIRS[21] onboard the Cassini spacecraft. Saturn's emitted power in some years of the Cassini epoch was discussed in a couple of studies[16,25]. However, this study is the first to discuss measurements of Saturn's emitted power during the complete Cassini epoch (2004–2017). The Cassini CIRS has three focal planes (FP1, FP3, and FP3). The wavenumber ranges of the three focal planes are 10-695 cm$^{-1}$ (14–1000 μm), 570–1125 cm$^{-1}$ (9-18 μm), and 1025–1430 cm$^{-1}$ (7–10 μm), respectively. The radiance recorded by FP1 is dominant in the total thermal radiance emitted from Saturn[16]. Fig. S1 shows the coverage of FP1-recorded thermal radiance in the two-dimensional domain of emission angle and latitude, which suggests that there are still observational gaps even for the best observations from Cassini. The least-squares fitting, which was applied in our previous study[16], is used to fill the observational gaps in the direction of emission angle for each latitude.

Figure S2 provides the complete distribution of radiance after filling the observational gaps. By integrating the radiance in the direction of emission angle, we obtained the meridional profiles of Saturn's emitted power in different years, which are displayed in Fig. 1 of the main text. The error bars depicted in Fig. 1, which illustrate the measurement uncertainties, are estimated by combining the CIRS data calibration errors with the uncertainties associated with filling the observational gaps[16]. Then, the meridional distribution of emitted power is averaged over latitude to obtain the global and hemispheric averages (Fig. 2). The estimates of uncertainties in the global and hemispheric averages are based on the error-bars shown in the meridional profiles of emitted power by applying the rule of error

propagation[16,46]. As discussed in the main text, the measurements of Saturn's emitted power are further extrapolated from the Cassini epoch (2004-2017) to the whole orbital period (1995–2025), which are shown in Figs. 3 and 4.

### Solar irradiance and the effects of Saturn's rings

The other energy component in Saturn's radiant energy budget, the absorbed solar energy, is primarily determined by the bolometric Bond albedo[1,2]. The bolometric Bond albedo is sometimes simply referred to as Bond albedo, which is further determined by the monochromatic Bond albedo. The monochromatic Bond albedo (i.e., Bond albedo at each wavelength)[11,26] is defined as the ratio of the absorbed solar irradiance to the incoming solar irradiance at each wavelength. For Saturn, the incoming solar irradiance to Saturn's atmosphere is modified by Saturn's rings. The solar irradiance and its modifications by the rings are discussed here.

The incoming solar irradiance at each wavelength, also known as the solar spectral irradiance (SSI), provides the reference for computing the monochromatic Bond albedo[11,26]. The SSI has temporal variations larger than 10% at some wavelengths[26,28]. Therefore, it is better to include the temporal variations of the SSI in the measurements of Saturn's monochromatic Bond albedo. The SSI during the Cassini epoch (2004–2017), constructed from multiple data sets, was provided in our previous studies of Titan and Enceladus[26,28]. The solar irradiance, which is an integration of the SSI over wavelength, is used in our investigations of Saturn's Bond albedo and the absorbed solar power.

For Saturn, there is difficulty in measuring the Bond albedo due to the modification of incoming solar irradiance at the top of the atmosphere by Saturn's spectacular rings. Saturn's rings can block solar irradiance to the atmosphere by casting shadows on it (i.e., ring-shadowing). In addition, Saturn's rings can scatter solar irradiance to the atmosphere (i.e., ring-scattering). Finally, the thermal emission from the rings (i.e., ring-emitting) also affects the radiant energy budget. The first two effects (ring-shadowing and ring-scattering) directly modify the solar irradiance at the top of the atmosphere. The third effect (ring-emitting), which is concentrated in the far-infrared wavelengths due to the cold temperature of the rings, does not significantly affect the solar irradiance in the visible and near-infrared wavelengths. However, this third effect should be considered in the analysis of Saturn's radiant energy budget. In this study, the first two effects (ring-shadowing and ring-scattering) are considered in the modification of solar irradiance at the top of Saturn's atmosphere and hence the measurements of Bond albedo. Then, we assume that the thermal emission from the rings to the atmosphere is reflected at a ratio identical to the Bond albedo measured in the visible and near-infrared wavelengths when considering the thermal emission in Saturn's radiant energy budget. This simple assumption probably overestimates the Bond albedo of the rings' thermal emission concentrating in the middle and long infrared wavelengths because the absorption and scattering of cloud particles are probably strong in these wavelengths. However, this assumption essentially does not affect our analysis of Saturn's radiant energy budget because the thermal emission from the rings is smaller by at least one order of magnitude compared to the scattering and blocking of the rings (Figs. S3 and S5).

To compute the three effects from the rings, a model of the rings was developed in previous studies[12–14]. The computation of ring-shadowing effect on an oblate planet, developed in a study[74], was included in a model of rings[12,14]. The equations for the ring-scattering effect were developed in another study[12], which were accounted for in the model of rings[12,14]. The model depends on the thermal and optical properties of Saturn's rings. Previous computations of the rings' effects[12–15] are based on the measurements of the rings' properties from the observations recorded by Pioneer[75] and Voyager spacecraft[76].

The thermal emission from the rings, which mainly depends on the temperature of the rings, has a much smaller effect than the shadowing and scattering effects. Therefore, we still use the temperatures of the rings retrieved from observations recorded by the Pioneer infrared radiometer[75]. On the other hand, the optical depth, which plays a critical role in computing the ring-shadowing and ring-scattering effects, has been significantly updated since the Pioneer/Voyager epochs. In particular, the in-orbit observations conducted by the Cassini spacecraft significantly improved our understanding of the rings' optical characteristics. Here, we use the optical characteristics of Saturn's rings from recent studies based on the Cassini observations[29,30] to compute the ring-shadowing and ring-scattering effects.

Figure S3 shows the meridional distributions of the three effects of the rings (i.e., shadowing, scattering, and emitting) at the top of Saturn's atmosphere from 1995 to 2025, a complete orbital period including the Cassini epoch. The corresponding solar longitude (Ls), which is defined as the longitude of the Sun on the sky in a Saturn-centered reference frame, is also shown in the figure. The seasons are defined as Ls = 0–90° for the spring of the NH (autumn of the SH), Ls = 90–180° for the summer of the NH (winter of the SH), Ls = 180–270° for the autumn of the NH (spring of the SH), and Ls = 270–360° for the winter of the NH (summer of the SH).

The total effect from ring-shadowing and ring-scattering is displayed in panel A of Fig. S4. The magnitude of the ring-shadowing effect is generally larger than that of the ring-scattering effect. Panel B of Fig. S4 shows the solar irradiance at the top of Saturn's atmosphere without the effects of the rings, which is constructed based on the time-varying SSI, Saturn's oblateness, the eccentricity of Saturn's orbit around the Sun, and Saturn's obliquity to its orbital plane. Panel C shows the modified solar irradiance at the top of the atmosphere by considering the total effect from the ring-shadowing and ring-scattering (panel A). This panel shows that the rings' effects significantly modify the solar irradiance at the top of Saturn's atmosphere. At some latitudes, the rings' effect is comparable to the original solar irradiance. It should be mentioned again that the third effect (i.e., ring-emitting) was not considered in the modified solar irradiance because it is concentrated in the far infrared. Such an effect is included in our analysis of Saturn's radiant energy budget.

Using the meridional distribution of solar irradiance shown in Figs. S3, we can calculate the global averages of the three effects from Saturn's rings (Fig. S5) by considering the oblate shape of the planet[16,77]. Figure S6 displays the global-average total effect of ring-shadowing and ring-scattering (panel A) and the global averages of the original and modified solar irradiance (panel B). The modified global-average solar irradiance shown in panel B serves as the reference to compute the Bond albedo with the reflected solar irradiance observed by the Cassini spacecraft. Additionally, we calculate the hemispheric averages of the three effects from the rings (panels A and B of Fig. S7) and the modified solar irradiance (panels C and D of Fig. S7). The hemispheric calculations will be used in the analysis of the absorbed solar power at the hemispheric scale (Fig. S25).

We also try to validate the computed effects of the rings. Among the three effects of the rings, ring-scattering and ring-emitting are difficult to observe directly. However, the shadows cast by the rings can be observed. We can validate the computation of the rings' model by comparing the observed shadows with the computed shadows. Figure S8 shows a global image of Saturn recorded by the imaging system of the Cassini spacecraft in May 2011 (for more details of the Cassini data, see the next section). The observational time (May 2011) corresponds to the spring of the NH with a sub-solar-latitude of 9.5°N, which means that the shadows cast by the rings appear in the SH (panel A). The sub-Cassini-latitude is very small (0.2°N), so the rings only block the Cassini view from a very narrow latitude band around the equator (see the thin line at the equator of the global images shown in panel A). Panel B of Fig. S8 shows the navigated latitudes for the image shown in panel A, while panel C only shows the latitudes covered by the rings' shadows. Panel D of Fig. S8 is the distribution of the ring-shadowing effect with latitude, which is outputted from the rings' model. The latitude range outputted from the model (3-19°S) is consistent with the observed latitude range shown in panel C. Furthermore, the rings' model also captures the Cassini division between the A ring and B ring, which is shown as a peak of the solar irradiance around 10°S in panel D.

## Observations of Saturn's full-disk reflectance

The ring-modified solar irradiance discussed in the previous section is an integration of SSI over wavelength, whereas the Cassini observations of Saturn's reflected solar irradiance were recorded at various wavelengths. Therefore, we need to convert the modified solar irradiance into modified SSI. The optical depths of the rings are generally greater than 1 in the visible and near-infrared wavelengths[29,30], indicating that ring-shadowing is independent of wavelength. This assertion is further supported by Cassini/ISS observations (Fig. S23). Additionally, ring-scattering does not significantly vary with wavelength in the visible and near-infrared spectra[78,79], which primarily contributes to solar flux. Therefore, we can use the ratio between the modified solar irradiance and the original solar irradiance to estimate the modified SSI. With the modified SSI, we require observations of the reflected SSI to calculate the monochromatic Bond albedo. These observations are mainly recorded by the Cassini spacecraft. Additionally, we use supplementary observations recorded by other observatories in our analysis of Saturn's Bond albedo. This section introduces these observations and datasets.

The Cassini spacecraft conducted long-term observations of Saturn from October 2004 to September 2017. In this study, we mainly analyze observations of Saturn's reflected solar radiance recorded by two Cassini instruments: the ISS[18] and the VIMS[22]. Compared to previous observations, the Cassini observations are better suited for measuring the Bond albedos and absorbed solar powers of planets and satellites. This is due to several advantages they offer, such as better coverage of wavelength and viewing geometry. These advantages have been previously discussed in our studies[11,26,28].

The characteristics of the ISS instrument and related data processing (e.g., calibration and navigation) have been described in previous studies[11,18,26,28,80,81]. In this study, we mainly examine the observations recorded by the ISS 12 filters, which include three ultraviolet filters (UV1 at 264 nm, UV2 at 306 nm, and UV3 at 343 nm), three methane-absorption filters (MT1 at 619 nm, MT2 at 728 nm, and MT3 at 890 nm), three continuum filters (CB1 at 635 nm, CB2 at 752 nm, and CB3 at 939 nm), and three color filters (BL1 at 463 nm, GRN at 568 nm, and RED at 647 nm)[18]. Figure S9 shows an example of calibrated ISS global images recorded by the RED filter. The calibrated radiance at each pixel of the global image is multiplied by the projected area of the pixel over Saturn's surface and then summed over all pixels in the disk to obtain the full-disk reflected spectral intensity[26]. The reference spectral intensity, based on the modified SSI, is multiplied by the total area of Saturn's disk to obtain the reference full-disk solar spectral radiance. Then, Saturn's full-disk reflectance can be computed by dividing the observed reflected spectral intensity by the reference solar spectral intensity[26].

Saturn's rings introduce difficulty in measuring the Bond albedo by modifying the incoming solar irradiance, as discussed above. Another difficulty is caused by the rings blocking Cassini's view of Saturn's atmosphere, especially when the spacecraft was away from the plane of the rings. Therefore, we carefully select global images that avoid serious obstruction of the Cassini view by the rings. Our tests suggest that if the sub-Cassini latitude is less than 3°, the observational gaps caused by Cassini's view being blocked by the rings can be filled using linear interpolation. Figure S10 shows an example of our selected

global images. Panel A shows the original global image, which was calibrated using the ISS calibrated software - the Cassini ISS CALibration (CISSCAL, version 4.3)[80,81]. The raw image was taken by Cassini in July 2010 (spring of the NH) with a sub-solar latitude of 5.1°N, so the shadows cast by the rings appeared in the SH (i.e., the lower blue belt shown in panel A). The Cassini recorded the image with a sub-Cassini latitude of 2.6°S, which means that the Cassini was under the plane of the rings at the observational time. Therefore, the Cassini view blocked by the rings takes place in the NH (i.e., the upper blue belt shown in panel A). Because of the small sub-Cassini latitude (<3°), the latitude belt blocked from the Cassini view is narrow. We can linearly interpolate the observed solar radiance in the neighboring latitudes to fill the observational gaps, as shown in panel B of Fig. S10.

Based on the selection criterion (i.e., sub-Cassini latitude <3°), we only found high-quality global images at 7 filters (RED, GRN, BL1, CB2, CB3, MT2, and MT3). For these 7 filters, very few global images were recorded at phase angles less than 20° due to the observational geometry of Cassini. At low phase angles, the Cassini spacecraft was generally close to Saturn, so only parts of Saturn's full disk were recorded. To increase the coverage of phase angles, we constructed Saturn's global images based on some quasi-simultaneous quarter images recorded by the ISS. Examples of making global images from quarter images taken in a $2 \times 2$ mosaic are shown in Fig. S11. The four quarter images were recorded by the Cassini ISS at a phase angle of 15.9°, so the corresponding global image can help fill the gap in phase angle. By searching the public Cassini ISS data, we found two groups of quasi-simultaneous images to make two global images for each of the three color filters (RED, GRN, and BL1). The phase angles for these global images change from 15° to 17°. We first calibrate the global images made from the quarter images and then compute Saturn's full-disk reflectance at the corresponding phase angles.

The Cassini ISS conducted observations of Saturn from 2004 to 2017. In principle, we could explore the temporal variations of Saturn's Bond albedo. However, the poor coverage of phase angles in Saturn's global images makes it difficult to compute the Bond albedo for each year of the Cassini period. As shown in Fig. S12, the ISS global observations have very sparse coverage of phase angle for most years of the Cassini period. Therefore, the Cassini's global observations in each year are not sufficient to measure Saturn's Bond albedo, which requires good coverage of phase angle. Nevertheless, we can investigate the temporal variations of Saturn's full-disk reflectance at these phase angles for which the ISS global observations are available at multiple times.

Panels A-C in Fig. S13 show global images captured by the ISS CB2 filter at different times but with the same phase angle of 62°. Panel D illustrates the corresponding full-disk reflectance, which suggests that Saturn's full-disk reflectance varied by approximately 1.1–2.6% during the Cassini mission. Figure S14 displays global images captured by the BL1 filter at different times but with the same phase angle of 62°. Panel B of Fig. S14 displays the bright cloud belt generated by the 2010 giant storm, allowing us to examine the effect of the 2010 giant storm on Saturn's full-disk reflectance. Panel D indicates that Saturn's full-disk reflectance increased by 2.9% in 2011 (panel A) than in 2007 and 2012 (panels B and C). The increase of full-disk reflectance in 2011 is due to the bright cloud belt generated by the 2010 giant storm. However, the increase in Saturn's full-disk reflectance caused by the 2010 giant storm (2.9%) is comparable to the temporal variations of Saturn's reflectance recorded by the CB2 observations (Fig. S13). Since the temporal variations of Saturn's full-disk reflectance (a couple of percentages) are lower than the measurement uncertainty of Saturn's Bond albedo (approximately 5%, see the uncertainty discussion in the following sections), they are not considered in this study.

Therefore, we combine all the ISS observations from the Cassini period (2004-2017) to explore the time-mean full-disk reflectance and hence the Bond albedo of Saturn. The ISS measurements of Saturn's full-disk reflectance based on all available global images at the seven filters (RED, GRN, BL1, CB2, CB3, MT2, and MT3) are shown in Fig. 5 of the main text. The global images created from quarter images taken with the RED, GRN, and BL1 filters (panels A-C) provide better coverage of the low phase angle range compared to the measurements obtained with other filters (panels D-G).

The Cassini ISS observations have the best coverage of phase angle, but they are limited in wavelength coverage. The Cassini spacecraft carried another instrument with much better wavelength coverage. The Cassini VIMS[22] is an imaging spectrometer that acquired images at 352 wavelengths between 350 nm and 5131 nm. Therefore, the VIMS observations can help extend the spectral coverage of the ISS observations. The processing of the VIMS data was described in previous studies[22,82–85]. Here, we used the Geological Survey Integrated Software for Imagers and Spectrometers (ISIS3) (https://isis.astrogeology.usgs.gov/7.0.0/UserStart/index.html) to calibrate the VIMS data. The VIMS visual (VIS) and infrared (IR) channels have spectral ranges of 350-1046 nm and 891-5131 nm, respectively. A study[83] suggested that scaling the IR to the VIS spectra around 980 nm provides good results for the continuity of the VIMS spectra over its overlapped wavelengths between the VIS and IR observations. Here, we followed the study to merge the VIS and IR spectra.

Unfortunately, the VIMS was unable to directly capture high-spatial-resolution global images of Saturn because it was limited to acquiring images with a maximum resolution of $64 \times 64$ pixels. As a result, the VIMS could only record global images at low spatial resolutions (worse than 2000 km/pixel). These low-spatial-resolution images did not provide sufficient detail to determine the effects of the rings. In addition, making global images from the VIMS regional images is difficult. Figure S15 shows one of the best attempts, but the combination of the nine VIMS images still does not cover the full disk of Saturn. In this study, the VIMS observations are mainly used to examine the spectral shape of Saturn's reflectance, which can help us extrapolate the phase functions from these wavelengths measured by the ISS to other wavelengths.

Based on the quasi-global image shown in Fig. S15, we can estimate the full-disk reflectance. By averaging Saturn's reflectance over the nine images shown in Fig. S15, we have the full-disk reflectance spectra (red line in Fig. S16). Additionally, we compare the global-average spectra with the reflectance spectra based on one image in Fig. S15 to examine whether the spatial coverage affects the shape of Saturn's reflectance spectra. The comparison in Fig. S16 suggests that the incomplete coverage of the full disk affects the magnitude of Saturn's reflectance, but it does not influence the spectral shape of Saturn's reflectance. Therefore, we can use the reflectance spectra averaging over the nine images to represent the spectral shape of Saturn's full-disk reflectance, even though the nine images do not completely cover the full disk of Saturn. It should be noted that the magnitude of the quasi-global spectra shown in Fig. S16 (red line) should be used with caution because (1) the quasi-global image does not cover the full disk of Saturn, and (2) the rings' effects were not considered in the data calibration. However, the spectral shape shown in Fig. S16 can help us fill the observational gaps in wavelength.

In addition to the ISS and VIMS, the Cassini spacecraft has another imaging system called the Ultraviolet Imaging Spectrograph Subsystem (UVIS)[86] that observed Saturn's reflected solar radiance. The UVIS has a wavelength range of 56 nm to 190 nm, which accounts for only 0.13% of the total solar irradiance. However, we could not find high-quality global reflectance spectra from the public database of the UVIS. Instead, we found available ultraviolet spectra of Saturn from other observations. These ultraviolet reflectance spectra and other observations, which are used to fill observational gaps and validate the Cassini measurements, will be introduced in the next section. The measurements based on observations by the ISS and VIMS (Fig. 5 and Fig. S16) have the shortest wavelengths at around 350 nm.

Additionally, we did not find good global data from the Cassini UVIS database for wavelengths shorter than 350 nm. Therefore, we searched for other datasets and published results of Saturn's full-disk reflectance in wavelengths shorter than 350 nm. There are numerous observations and studies of Saturn's reflectance in different wavelengths and times. Here, we focus on the best full-disk observations and studies. The data recorded by the European Southern Observatory (ESO)[45,53], located in La Silla, Chile, is among the best observations. The ESO observations were conducted by the Boller and Chivens spectrograph, which was mounted on a 1.52-m telescope. The spectrograph recorded Saturn's full-disk observations in 1993 and 1995 with a spectral range of 300-1050 nm and a spectral resolution of 0.4 nm.

Compared to the 1993 observations[53], the 1995 observations[45] are much better suited for measuring Saturn's full-disk reflectance. Firstly, the 1995 observations were conducted closer to the equinox, with a small sub-solar latitude of 2.0°N, while the 1993 observations were taken away from the equinox, and had a much larger sub-solar latitude of 11.9°N. Therefore, the shadows cast on Saturn's atmosphere by the rings are much larger in 1993 than in 1995. Secondly, the inclination of the rings with respect to Earth is much smaller for the 1995 observations (0.6°) than for the 1993 observations (10.9°), suggesting that the rings block much smaller parts of Saturn's full-disk from the ESO view in 1995 than in 1993. In addition, there are several other improvements in the analysis of the 1995 observations[45]. Therefore, the full-disk spectra based on the 1995 observations are used in this study (Fig. S17). The 1995 observations of Saturn's full-disk reflectance had a phase angle of 5.7°, making them helpful for filling the observational gaps in phase angle for the Cassini ISS global images.

The wavelengths covered by the ESO observations in 1995 range from 300 nm to 1050 nm, with a very high spectral resolution of 0.4 nm. The ESO observations do not cover wavelengths shorter than 300 nm. The International Ultraviolet Explorer (IUE)[51] observed Saturn in 1978-80 with a phase angle close to 0°, covering the spectral range of 120-194 nm with varying spectral resolutions from 0.1 nm to 1.2 nm. In addition, the Aerobee Rocket[52] recorded Saturn's full-disk reflectance in 1964 with a phase angle of 0.89° across four wavelengths from ultraviolet to visible (i.e., 245 nm, 280 nm, 295 nm, and 353 nm). Both spectra are displayed in Fig. S17 as well.

The observations other than the Cassini observations not only help us fill observational gaps in phase angle and wavelength, but also provide independent datasets to validate the Cassini measurements. Figure S18 shows a comparison of Saturn's reflectance spectra between the VIMS measurements and the ESO analyzes[45], which suggests consistency in the spectra shape between them over their overlapping wavelengths (350-1050 nm). It should be mentioned that the VIMS spectra do not resolve some fine structures shown in the ESO spectra because the spectral resolution is much better for the ESO observations (0.4 nm) than for the VIMS observations (4-24 nm). The magnitude of the spectra is different between the VIMS measurements and the ESO analysis, probably because (1) the two spectra were recorded at different phase angles; and (2) the magnitude of the VIMS spectra should be used with caution as discussed above. It is also worth mentioning that the ESO spectra were conducted closer to the equinox, so the effects of the rings are relatively small.

## Measurements of Saturn's Bond albedo

As discussed in the main text, the Cassini/ISS measurements are combined with the ESO observations[45] to investigate the phase functions of Saturn's full-disk reflectance at the seven filters of the ISS (i.e., GRN at 568 nm, RED at 647 nm, BL1 at 463 nm, CB2 at 752 nm, CB3 at 939 nm, MT2 at 728 nm, and MT3 at 890 nm). A polynomial function is used to fit the phase curve and hence fill the observational gaps in phase angle. Figure 7 in the main text displays the fitting results for the data recorded by the seven filters (wavelengths) of the Cassini ISS,

which suggests that the polynomial function works well for fitting Saturn's full-disk reflectance. The fitting results are used to fill the observational gaps in phase angle. The corresponding fitting residuals can be used to evaluate the fitting quality, which are displayed in Fig. S19. This figure indicates that the residual ratios are less than 10% for most phase angles smaller than 100°, except for some points in the fitting of MT2 and MT3 data (see panels F and G in Fig. S19). Therefore, the fitting works well for most of the ISS data. The relatively large ratios in MT2/3 and at large phase angles (>100°) are caused by the small full-disk reflectance in those areas, which do not significantly contribute to the measurements of Saturn's Bond albedo. It should be mentioned that the uncertainty related to the fitting residuals at all phase angles and all filters is considered in our analysis of measurement uncertainties.

The fitting presented in Fig. S23 generates complete phase functions at the ISS seven filters/wavelengths. Here, we discuss how to interpolate/extrapolate the complete phase functions from the ISS seven wavelengths to the entire spectral range under investigation. The ESO observations[45] provide continuous reflectance spectra from 300 nm to 1050 nm (Fig. S17). By combining the complete phase functions at the ISS seven wavelengths with the available ESO observations in other wavelengths, we can fill in the observational gaps in wavelength. For the wavelength gaps from 463 nm (the shortest wavelength of the ISS seven filters) to 939 nm (the longest wavelength of the seven filters), we first linearly interpolate the complete phase functions from the seven filters to other intermediate wavelengths. We then use the ESO spectra to adjust the interpolated phase functions. Figure S20 shows an example of how we obtained the phase function at 500 nm, which is between the ISS BL1 filter (463 nm) and GRN filter (568 nm). We have the complete phase function at 463 nm and 568 nm for the two wavelengths on either side of 500 nm. We first linearly interpolate the complete phase functions at 463 nm (red solid line) and 568 nm (blue solid line) to obtain the phase function at 500 nm (black dashed line). We then use the ESO-observed reflectance at 500 nm with a phase angle of 5.7° to adjust the interpolated phase function. The ratio of full-disk reflectance at the phase angle of 5.7° between the interpolated results (black dashed line) and the ESO observations is used to adjust the interpolated phase function (black dashed line in Fig. S20) to the final phase function for the wavelength 500 nm (black solid line in Fig. S20).

For wavelengths less than 463 nm, we scale the phase function at 463 nm to match the observed reflectance spectra at specific phase angles. In other words, we use the ESO spectra[45] from 300 nm to 463 nm (5.7° phase angle), the Aerobee spectra[52] from 245 nm to 300 nm (0.89° phase angle), and the IUE spectra[51] from 120 nm to 194 nm (0° phase angle), which are shown in Fig. S17, to fill in the observational gaps in this range. For the wavelength gaps between the IUE and Aerobee coverage (195–244 nm, see Fig. S17), we interpolate the phase functions at 194 nm and 245 nm to fill the phase functions from 195 nm to 244 nm.

For wavelengths longer than 939 nm but shorter than the longest wavelength observed by VIMS (5131 nm), we use both the phase functions from the CB3 filter (939 nm) and MT3 filter (890 nm) to extrapolate the phase functions at wavelengths longer than 939 nm. The Cassini ISS observations and fitting results (Fig. S23) demonstrate that the phase functions differ between the continuum bands (e.g., CB2 and CB3) and the methane-absorption bands (e.g., MT2 and MT3), so we use the phase functions at the MT3 and CB3 wavelengths to scale these wavelengths with and without methane absorption, respectively. We use the methane-absorption band around 1400 nm (Fig. S16) as an example to demonstrate how to scale the phase angle at MT3 (890 nm) to obtain the phase function at the wavelength of 1400 nm. Firstly, we calculate the ratio of full-disk reflectance at a phase angle of 11.5° between 1400 nm and 890 nm based on the spectra recorded by the Cassini/VIMS (Fig. S16). Then, we scale the complete phase function at

890 nm by this ratio to obtain the phase function at the wavelength of 1400 nm.

After filling the observational gaps in wavelength, we have the full-disk reflectance in the domain of phase angle and wavelength, which is shown in Fig. 8 of the main text. Based on Fig. 8, we can integrate the full-disk reflectance over phase angle to get the monochromatic Bond albedo (i.e., Bond albedo at each wavelength)[26], which is shown in Fig. S21. The wavelength-averaged Bond albedo, also known as the Bond albedo, is calculated by weighting the monochromatic Bond albedo with the SSI[26]. Based on the monochromatic Bond albedo displayed in Fig. S21 and the SSI provided in our previous studies[26,28], we have Saturn's Bond albedo as $0.41 \pm 0.02$.

Now, we discuss the uncertainties in the measurements of Saturn's Bond albedo, which affect the analysis of absorbed energy and the energy budget. Saturn's Bond albedo depends on the monochromatic Bond albedo, which is determined by the full-disk reflected solar radiance at different phase angles and wavelengths. The uncertainties in the measurements of Saturn's monochromatic Bond albedo mainly arise from uncertainties in measuring the reflected solar radiance at different wavelengths and phase angles. We organize the uncertainty sources in the measurements of monochromatic Bond albedo into three categories: (1) noise related to calibrating the observational datasets; (2) uncertainty in estimating the effects of the rings; and (3) uncertainty related to filling observational gaps.

In this study, Saturn's full-disk reflectance is mainly determined by the Cassini ISS and VIMS observations. We first discuss the noise related to the calibration of the ISS and VIMS data. We used the latest version of the Cassini ISS CALibration software[61] to calibrate the ISS data. The calibration errors have been discussed in previous studies[26,80,81], which are ~ 5% of the absolute calibrated radiance.

For Cassini VIMS data, the noise after calibration process[22,83,85] is also on the order of 5% of the calibrated radiance[65]. Our computation of Saturn's Bond albedo also uses observations from other observatories. For simplicity, we assume that the calibration uncertainties of the other datasets have a similar magnitude to that of the calibration noise in the Cassini datasets. It should be emphasized that the calibration noise is not systematic. The uncertainties caused by the calibration noise become much smaller than 5% when the recorded radiance is integrated over wavelength and phase angle for computing the Bond albedo, because the noises at different wavelengths and phase angles cancel each other out.

The solar irradiance at the top of Saturn's atmosphere, which is modified by the two effects of the rings (i.e., ring-shadowing and ring-scattering) (Figs. S6 and S7), is used as the reference for computing the full-disk reflectance and hence Bond albedo. Estimating the uncertainty in computing the two effects of the rings is difficult because they involve multiple processes (e.g., blocking and scattering), optical characteristics of the rings, and the geometry model. Here, we use the temporal characteristics of the two effects to estimate the uncertainty in the computed effects of the rings. Based on the combined effect of ring-shadowing and ring-scattering (panel A of Fig. S6), we first compute the standard deviation of the temporal variations of the combined effect. Then, such a standard deviation is used to represent the uncertainty. The ratio between the standard deviation and the mean value of the modified solar irradiance is approximately 3%. Such a ratio is used to estimate the uncertainty of the modified solar irradiance, which is further accounted for in our analysis of the uncertainty of the measurements of Saturn's Bond albedo.

For the uncertainties related to filling observational gaps in phase angle and wavelength, we use fitting residuals to examine them. The fitting residuals shown in Fig. S19 are used to estimate the uncertainty related to filling observational gaps in phase angle. The basic idea is that we use the fitting residuals, which exist in the phase angles with available observations, to estimate the uncertainties in the observational gaps. The fitting residuals can also be applied to estimate the

uncertainties related to filling observational gaps in wavelength. Let us take the wavelength 500 nm as an example. First, we can use the above method to estimate the uncertainties related to filling observational gaps in phase angle at the two observed wavelengths (463 nm and 568 nm), which are on two sides of the wavelength 500 nm. Then we linearly interpolate the estimated uncertainties at 463 nm and 568 nm to the uncertainty at 500 nm and use it to estimate the uncertainty at 500 nm. To evaluate the uncertainties at wavelengths outside of the seven wavelengths, we simply extrapolate the uncertainties at the seven wavelengths to estimate the uncertainties at these wavelengths.

To obtain the total uncertainty of the monochromatic Bond albedo of Saturn, we combine three sources of uncertainty: calibration noise, errors in computing ring effects, and imperfect fitting. We follow the rule of error propagation of addition, as described in a study[46]. Saturn's Bond albedo is equivalent to the sum of the monochromatic Bond albedos at different wavelengths[26]. Therefore, we can estimate the uncertainty of Saturn's Bond albedo by applying the rule of error propagation of addition again to the uncertainties of the monochromatic Bond albedos at different wavelengths.

## Computations of Saturn's absorbed solar power

Based on investigations of the rings-modified solar flux at Saturn (Figs. S6 and S7) and measurements of the Bond albedo, we can compute the hemispheric and global averages of Saturn's absorbed solar power during the Cassini epoch. We first investigate potential temporal fluctuations in Saturn's Bond albedo, both globally and hemispherically.

Saturn's brightness (i.e., reflected radiance) and its related optical characteristics display temporal variations[87–90]. The Cassini observations also suggest that Saturn's brightness changed with time during the Cassini epoch (panels A, B, C of Figs. S13 and S14) even though its full-disk reflectance did not significantly change with time (panel D of Figs. S13 and S14). The seasonal variations of solar irradiance at Saturn (Figs. S6 and S7), which are determined by both the Sun-Saturn distance and ring effects, significantly affect the temporal variations of Saturn's brightness. The temporal variations of cloud activities and banded structures also influence the seasonal variations of Saturn's brightness[88–90]. Compared to the significant temporal variations in Saturn's brightness, Saturn's global and hemispheric reflectance, as well as its Bond albedo, exhibit relatively minor temporal fluctuations. The Cassini ISS observations reveal that Saturn's full-disk reflectance varied about 1.1-2.9% during the Cassini epoch (panel D of Figs. S13 and S14), which is smaller than the ratio between the measurement uncertainty and the Bond albedo (approximately 5%). Consequently, the temporal changes in Saturn's full-disk reflectance during the Cassini epoch lack statistical significance. Additionally, previous studies[45,53,54] also suggest that Saturn's full-disk reflectance does not significantly vary at the longer time scales. Finally, the regional modifications of Saturn's brightness do not significantly change the global and hemispheric albedos. For example, Fig. S14 suggests that the bright cloud band generated by the 2010 giant storm increased Saturn's reflectance by 2.9% and 5.8% for the global and hemispheric reflectance, respectively. This implies that the modifications of global and hemispheric Bond albedo by the strongest atmospheric events (e.g., giant storms) are still smaller or around the measurement uncertainty of Saturn's Bond albedo. Therefore, we did not consider the temporal variations of the global and hemispheric albedos in the analysis of Saturn's absorbed solar power during the complete orbital period, including the Cassini epoch (1995-2025).

Saturn's brightness also displays hemispheric asymmetry, especially when Saturn moves away from its equinox points in its orbit around the Sun[88,90–93]. Figures S13 and S14 reveal the difference in Saturn's brightness between the NH and SH at two wavelengths (463 nm and 752 nm). The hemisphere directly facing the sun (the subsolar hemisphere) receives more solar irradiance than the other

hemisphere. Saturn's rings further strengthen the hemispheric asymmetry by scattering solar irradiance to the subsolar hemisphere and blocking solar irradiance to the other hemisphere (Fig. S7). Correspondingly, the subsolar hemisphere has greater brightness than the other hemisphere, as shown in Figs. S13 and S14. The hemispheric difference in cloud opacity also contributes to the brightness contrast between the two hemispheres[88,90–95].

The hemispheric asymmetry of Saturn's brightness does not necessarily suggest the hemispheric asymmetry of reflectance and Bond albedo. Reflectance and Bond albedo are primarily determined by the ratio between incident solar radiance and reflected solar radiance. While incident solar radiance and reflected solar radiance (i.e., brightness) can vary between hemispheres, it is possible for the ratio between them to remain constant from one hemisphere to the other. Here, we examine possible differences in Saturn's reflectance and, hence, Bond albedo between the two hemispheres.

The methodology used for computing full-disk Bond albedo[1,2,11,26,28] does not work for investigating Bond albedo at regional and hemispheric scales. For regional and hemispheric Bond albedos, the Bidirectional Reflectance Distribution Function[96] should be considered. Here, we select the Cassini ISS observations around the NH spring equinox (i.e., August 2009) with the sub-Cassini-latitudes close to the equatorial plane of Saturn. Such a selection minimizes the effects of the rings on Saturn's atmosphere and the blocking of the Cassini view by the rings. Additionally, the two hemispheres can be compared with the basically same viewing geometry. We searched for such global observations from the entire ISS dataset but, unfortunately, did not find any observations at low phase angles (i.e., phase angles <100°). However, we did find some observations at high phase angles, from 102° to 120°, at the six filters (RED, GRN, BL1, CB2, MT2, and MT3).

Figure S23 shows examples of high-phase-angle global images taken at times around the spring equinox. Banded structures appear in these high-phase-angle images at different filters, particularly in two methane-absorption bands (MT2 and MT3, see panels E and F in Fig. S23). In addition, Fig. S23 shows the roughly symmetrical pattern of reflected radiance between the two hemispheres. Figure S24 displays a comparison of the hemispheric reflectance between the two hemispheres, which confirms that the hemispheric reflectance is generally the same between the two hemispheres. Figure S24 also shows that the NH reflectance is slightly larger than the SH reflectance at all filters except for CB2. The CB2 measurements suggest that the NH reflectance is larger than the SH reflectance at some phase angles, but smaller than the SH reflectance at other phase angles. However, the ratio of the difference in reflectance between the two hemispheres to the hemispheric reflectance is smaller than 3%. This ratio is also smaller than the ratio between the measurement uncertainty and Saturn's Bond albedo, which is approximately 5%. If the basic consistency between the two hemispheres revealed in the limited wavelengths and phase angles (Fig. S24) holds for other wavelengths and phase angles, we can assume that the Bond albedo probably does not significantly change from one hemisphere to the other during the complete orbital period, including the Cassini epoch.

After discussing the spatio-temporal variability of Saturn's Bond albedo, we can use the measured full-disk Bond albedo to calculate the global and hemispheric averages of absorbed power. To obtain the total incoming power, we add the thermal emission from the rings (Fig. S5) to the modified solar irradiance (Fig. S6), despite the rings' thermal emission being much smaller than the modified solar irradiance on a global scale. Additionally, we assume that Saturn's full-disk Bond albedo remains constant throughout the seasons, as previously discussed. Consequently, we combine the seasonally constant Bond albedo with the seasonally variable incoming power to calculate the global-average absorbed power during the Cassini epoch and the entire orbital period (1995-2025), as shown in panel A of Fig. S25. The error bars in this panel primarily

stem from the uncertainties in measuring Saturn's Bond albedo. It should be noted that there are uncertainties in the total incoming power as well. While the known Sun-Saturn distance primarily determines the total incoming power, the effects of the rings introduce uncertainties. Nevertheless, the effects of the rings have already been factored into the uncertainty analysis of Saturn's full-disk Bond albedo. Therefore, we do not need to account for uncertainties in the total incoming power when calculating the absorbed power. The comparison between the global-average absorbed power and emitted power is used to determine Saturn's global cooling and internal heat (see Fig. 9 in the main text).

Figures S23 and S24 suggest that the Bond albedo does not vary between hemispheres. Therefore, the global Bond albedo can also be used to calculate the hemispheric averages of absorbed power for the Cassini epoch and the entire orbital period. Panel B of Fig. S25 shows that the seasonal variations of absorbed power are much stronger in the hemispheric averages than in the global average. The solar irradiance and the related absorbed power of the two hemispheres have opposite phases, which partially cancels out the seasonal variations at the global scale. This helps to explain why the magnitude of seasonal variations is smaller in the global analysis than in the hemispheric analysis. The hemispheric averages of the absorbed power are combined with the hemispheric averages of the emitted power (Figs. 3 and 4) and the estimated internal heat from the global analysis to determine Saturn's energy budgets at the hemispheric scale, which is discussed in Fig. 10 of the main text.

## Data availability
The Cassini raw data used in this study are publicly available from NASA Planetary Data System at https://pds-atmospheres.nmsu.edu/data_and_services/atmospheres_data/Cassini/Cassini.html. Specifically, the Cassini data sets of the Composite Infrared Spectrometer (CIRS), Imaging Science Sub-system (ISS), and Visual and Infrared Mapping Spectrometer (VIMS), which are analyzed in this study, can be downloaded from https://pds-atmospheres.nmsu.edu/data_and_services/atmospheres_data/Cassini/inst-cirs.html, https://pds-atmospheres.nmsu.edu/data_and_services/atmospheres_data/Cassini/inst-iss.html, and https://pds-atmospheres.nmsu.edu/data_and_services/atmospheres_data/Cassini/inst-vims.html, respectively. The datasets generated during and/or analyzed during the current study are available from the corresponding author on request.

## Code availability
The Geological Survey Integrated Software for Imagers and Spectrometers (ISIS3), which was used to process the Cassini data, is available on https://isis.astrogeology.usgs.gov/7.0.0/UserStart/index.html. The software Matlab was used to further process and analyze the Cassini data. The Matlab codes of rings' effects, which were used for computing Saturn's albedo, are direct implementations of the published model of rings[12–14]. Additionally, the Matlab codes of analyzing Saturn's radiant energy budget are direct implementations of published methods[11,16,26].

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

## Acknowledgements

We appreciate the invaluable help provided by Dr. Christopher D. Barnet, who shared his notes for computing the effects of Saturn's rings on the radiant energy budget. We also extend our gratitude to Erich Karkoschka for providing Saturn's data based on observations from the European Southern Observatory. Furthermore, we are grateful to the Cassini CIRS, ISS, and VIMS teams for recording the raw data sets. Finally, Liming Li acknowledges the support received from the NASA ROSES Cassini Data Analysis Program (NNX15AI85G, LL) and the Planetary Data Archiving, Restoration, and Tools Program (NNX16AG46G, LL).

## Author contributions

L.L., X.W., and X.J. designed the research, conducted the data analysis, and wrote the manuscript. P.F., R.W., and C.N. provided assistance in processing and analyzing the data from the Cassini VIMS, ISS, and CIRS, respectively. L.G., T.G., and R.A. provided constructive suggestions during group meetings. J.C. provided the updated Cassini data on the optical depth of Saturn's rings. S.P. and A.Sanchez-Lavega shared their ring model to calculate the effects of the rings on Saturn's radiant energy budget. T.G., M.H., M.K., A.M., A.Simon, D.W, and X.Z. participated in analyzing the results and revising the manuscript.

## Competing interests

The authors declare no competing interests.
