## [Peer Review File · Nature Communications]

REVIEWER COMMENTS

Reviewer #1 (Remarks to the Author):

This is a very interesting paper and represents an impressive amount of work in analysing a huge quantity of data. New estimates are presented of the Bond Albedo of Saturn and new insights gained into the radiative balance in this planetary atmosphere.

Although the work is substantive and important, I found it rather difficult to follow. Because of the format of Nature publications, the main paper has to be concise with details of the analysis presented instead in the Methods section, or in supplementary information. In this paper, I felt that this division had gone perhaps too far, with the main paper reduced almost to an extended abstract, and most of the scientific details of the paper contained substantially in the supplementary information paper. As a result, in order to make sense of what the authors have done the reader has to read both papers in parallel, which is difficult and time-consuming.

As to the work itself, it is mostly very impressive, but I felt there were a number of assumptions made that were not fully defended. That doesn't mean that I think these assumptions are wrong, it's just that I felt the assumptions needed to be better justified and evidence provided.

Particular worries I have are:

1. The Cassini CIRS data cover only just over a 1/3 of one Saturn year, so most of the thermal component is estimated from an extrapolation, which is not, in my view, very well explained or justified.
2. A very different estimate of the Bond Albedo is derived from the Cassini data to that resulting after the Voyager flybys. The authors suggest a reason why this might be different, but could, I think, have done more in demonstrating their hypothesis to be true. To be specific, the authors state that the Voyager estimate comes from analysing latitudes 11S to 32S, which is a dark latitude band, while their value is a true global average. Why couldn't the authors have demonstrated that if they also only considered 11S to 32S in their data that they'd get the same answer as the previous estimate?
3. I have a minor concern with the fitted phase functions to some Cassini/ISS wavelengths, which I detail later.

My remaining comments are mostly minor and are listed below for the main paper and the supplemental information paper.

Main article

Abstract: Many 'probables' and 'possibles' are listed in this abstract. Is this perhaps a bit too tentative? I think maybe it would be better to mention those things that the authors are confident of with a bit more certainty.

Page 3, line 59: delete 'extremely'?

Page 3, line 68: delete 'of'

Page 3, line 69: Sentence 'Such investigation is further used to re-examine the internal heat.' This sentence reads awkwardly and could be reworded.

Page 4, line 82: The Cassini CIRS data cover just 38% of the Saturn year and so much of these results depend on how these data were extrapolated to cover the rest of the year. But the details of this crucial extrapolation are buried within the extensive supplementary paper. I think details of this extrapolation should be included in the main paper.

Page 4, lines 88 – 93. An awful lot of the details of this study are in the supplemental material, which makes this really two papers in parallel, and thus rather hard to follow.

Page 4, line 95. The 'significant increase in emitted power in the middle latitudes of the NH from 2010 to 2011' is hard to see (in my opinion) from Figure 1. A time series of the emitted power at NH mid-latitudes would show this better. A good figure to show this is Fig. S3 in the supplemental information.

Page 5, line 109. Is it possible that the 'significant' difference between the Voyager and Cassini observations are due to systematic offsets between Cassini and Voyager, rather than 'climate change'? How do we know that the two datasets are directly comparable? I think some justification is needed.

Page 5, lines 118-121. The authors say that the difference between their derived Bond albedo than that deriving from Voyager observations may arise from the Voyager value coming only from observations from 11S to 32S. Couldn't this explanation be verified immediately by the authors reanalysing their Cassini data using the 11-32S latitudes only? Then, the authors would be able to compare directly with the Voyager estimate and presumably get the same answer, assuming this hypothesis is correct.

Page 6, line 125. Change 'changes' to 'change'

Page 6, line 133. The fact that Saturn emits more than it absorbs from the Sun was already very well known from Voyager times. Perhaps add a note to this effect here?

Page 6, lines 133-135. The use the word 'deficit' and then negative values and words such as maximum and minimum is confusing. The deficit, to me, implies a negative value. Hence, I would write this as, 'the radiant energy deficit (i.e., outgoing - incoming) changed from a minimum value of 2.63 W/m² in 2003 to a maximum value of 3.05 W/m² in 2013.

Page 6, line 137. Which 'current models and theories' are these? Add some references here, please.

Page 6, line 148. The emitted power curve is mostly extrapolated from the CIRS observations. In fact, the CIRS observations account for only 38% of the whole Saturn year. Hence, a lot of this paper's conclusions depend on the reliability of this extrapolation. I think this should be discussed more.

Page 7, line 155. 'explored in previous studies'. Again, can you add references please?

Page 7, line 156. 'led to a significant issue'. Can you be more precise - what 'significant issue' is this?

Page 7, line 158. 'covered three of Saturn's seasons'. I think this is a bit misleading. The observations in Fig 3C cover 2006 - 2017, i.e., 11 years. Compared with a Saturn year of 29.4 years. That's 38% of the year, which is hardly three seasons. It's not even two! I guess you could argue that you got all of southern spring, and small parts of the seasons either side, but that's not quite what is implied by this statement.

Page 7, line 167. I'm not sure CIRS captured the dominant part of the cycle. You have extrapolated the outgoing radiation either side of the Cassini Epoch so it's hardly surprising that the two time-ranges give similar values.

Page 7, line 168. 'This is further supported...' Aren't you saying the same thing again here?

Page 10, line 248. This is not really a very great difference between the obliquities of Saturn and Neptune.

Page 10, line 249. I don't agree. If Neptune's eccentricity is small then there shouldn't really be much difference between the balance in the northern and summer hemispheres.

Methods. This is a very short methods section, mostly referring the reader to other papers. I would have thought much of the information now in the supplementary paper could go here.

Page 14, line 327. Fig. 1 Caption. Change 'are' to 'is'

Page 14, line 331. Fig. 1 Caption. The 'dramatic increase of emitted power in the NH between 2010 and 2011' is hard to see on this plot as it is difficult to pick out the individual lines. It doesn't look any more significant than other inter-annual changes. Fig. S3 is better for this.

Page 15, line 342. Fig. 2 Caption. These data are from Cassini SSI and VIMS. I think the caption should state this explicitly.

Page 16, line 357. Fig. 3 caption. The black and magenta lines in Fig 3C are explained here, but I think it would be neater and easier to follow if these labels could be marked on the figure itself in some way.

Page 17, line 362. Fig. 3 caption. What is the evidence that the Bond Albedo is constant with seasons? This is discussed in the manuscript, but this statement in the caption should be qualified.

Page 17, line 368. Fig. 3 caption. 'extrapolation to the entire orbital period are discussed in the first section of SI.' This matter of how the CIRS data were extrapolated to other seasons seems really central and important to me, so I'm puzzled that it's been relegated to the supplementary information.

Page 17, line 371. Fig. 3 caption. Briefly summarise how these data have been 'extrapolated'.

Page 18, line 392. Add ', calculated as described in the text' after 'internal heat flux'.

Page 18, line 394. Add '(extrapolated as described in the supplementary material)' after 'extrapolated results'.

Supplementary Article

Page 3, Fig. S3. Here, the sharp increase from 2009 to 2011 for the northern hemisphere is obvious. Please, refer to this figure from main text when you discuss this effect.

Page 5, Fig. S5. The extrapolated curve in Panel B has a value of 4.85 in 1996, but the same curve has a value of 4.9 in 1996 in Panel C. How can this be?

Page 5, Figs S5A-C. How can a sine wave fitted to the modified data in Panel B match the original data in panel C? You must have modified the fit after 2017 in some way. The fact that these plots use different y-scales is not helpful. Something is missing here.

Page 6, lines 158-161. How can this be right? Saturn has generally stronger absorption bands in mid-IR, so more light would be absorbed before reflection. Also, the scattering of cloud particles is less effective at long wavelengths. Hence, assuming the albedo is the same at thermal IR as it is in the near-IR is a BIG assumption. Is there no way of computing an effective albedo at longer wavelengths? What effect might errors in this assumption make?

Page 6, line 175. Change 'on-orbit' to 'in-orbit'?

Page 7, line 199. Diagram (Fig S6) covers 1996 to 2025, not 1995 – 2025.

Page 8, line 234. Change 'with considering' to 'including'.

Page 11, Fig. S11. Can't you take the image A and roughly compute the reduction in solar irradiance caused by the ring shadow? You could assume N/S hemispheric asymmetry and then calculate how much the expected reflection has been reduced by the shadow as a function of wavelength and then compare directly with Panel D.

Page 12, line 307. Is it safe to assume that 'the effects of ring-shadowing and ring-scattering do not vary significantly with wavelength'? What is the evidence that this assumption is sensible?

Page 12, line 323. Fig. S12 Caption. What ISS wavelength is this?

Page 13, line 358. Fig. S13 Caption. Again, what ISS wavelength is this?

Page 13, line 366. Change 'built Saturn's' to 'constructed'

Page 19, line 498. Please give more details on what these ESO observations were, and which instrument/telescope they were made with. It's not enough, I think, to just give a reference and expect the reader to figure this out for themselves.

Page 20, Figure S21. Can't these be combined into a single figure? It would make it easier to quantitatively compare the data. Also, in terms of energy balance, what really matters is the product of the reflectivity and incident solar irradiance. which will underline the dominance of the reflectivity at green wavelengths over all others. I suggest changing this figure to one with two panels - one showing the reflectivity from all three instruments in the same plot, and one showing the same data, but multiplied by the solar irradiance.

Page 20, line 515. 'European Southern Observatory (ESO)' Again, what instrument is this? What telescope, in fact?!

Page 20, line 531. 'ESO observations'. Again, you need to define what instrument this was. And what telescope. It's presumably VLT but this should be stated here - the reader should not have to chase after references to learn this.

Page 22, Figure S23. How realistic are the forward-scattering peaks in this plot, especially in panels F and G? If the fitting model was physically based then we might be happy, but as it's just a 4th-order polynomial there's no physical basis for this as far as I can see.

Page 22, line 563. 'polynomial function works well' Well it obviously fits the data well, but how trustworthy is the extrapolation from a simple 4th-order polynomial like this?

Page 22, line 568. The data sort of show the forward scattering properties described. However, the fitted curves are not, in my view, demonstrated to be reliable.

Page 24, lines 607 – 619. I'm a little uneasy about this. As I understand it the authors have used the phase functions at CM3 and MT3 and applied these to longer wavelengths to extrapolate the VIMS observations. How much methane absorption was necessary to switch between the two phase functions? What about wavelengths where methane absorption is medium? Was it a simple binary switch between phase function depending on wavelength, or were the phase functions interpolated between in some way?

Page 24, Fig. S25. What are these panels, exactly? Are these from VIMS data? Or do they also include the SSI and 'ESO' data? For the geometric albedo, what were the observing conditions of what I assume is a VIMS spectrum (i.e., what at the solar zenith angle, viewing zenith angle, and phase angle)? For Panel B, please define what a phase integral is. Is this panel attempting to show the transition between the CM3 and MT3 phase functions?

Page 25, lines 631-632. As I noted earlier, can't you calculate the bond albedo from your data, limiting to these latitude bands, and verify that this accounts for the discrepancy between these estimates of the Bond Albedo?

Page 26, lines 667-668. Can this 'cancelling out' of the noises at different wavelengths and phase angles be validated or estimated?

Page 26, line 699. Why is this reference spelt out in terms of the authors, while all others are simply referred to by their numbers?

Reviewer #2 (Remarks to the Author):

The authors investigated Saturn's radiant energy budget and its seasonal variability mainly by using the multi-instrument long-term observations from Cassini (spanning from 2004 to 2017, partially covering three of Saturn's seasons). Their findings reveal that the energy budget exhibits significant dynamical imbalances, seasonal variations of the planetary cooling, and higher Bond albedo and internal heat flux values than previous estimations.

These results are interesting, important and hold significance in the field, considering that the majority of current evolutionary and atmospheric models for giant planets operate under the assumption of a globally balanced energy budget across all time scales. This widely accepted assumption within the scientific community is primarily attributed to the lack of appropriate long-term observations, and it facilitates the development of complex models. Additionally, the radiant energy budget holds importance for internal heat estimations.

The authors conducted a careful analysis of archival Cassini/CIRS, ISS, and VIMS data, presenting it in a relative sufficient detail. They employed appropriate methods for estimating both emitted and absorbed powers, as described in previous studies. As an additional aspect beyond the state-of-the-art, the authors introduced the computations of the effects of the Saturn rings on Bond albedo estimation.

Suggested improvements/comments:

Line 33: Similar to Jupiter and Uranus, it is expected that Saturn's overall energy budget is not in a stable state, which is unsurprising. Various factors may influence this equilibrium, such as the direct absorption of incoming solar irradiance by the atmosphere, limited thermal radiation from deeper levels, and the presence of a heat source. Please rephrase the sentence accordingly.

Line 55-57: Please add references to the sentence.

Lines 83-84: Please see comment in the SI below regarding lines 147-172.

Lines 107-109: Temporal variability is a plausible factor, though it remains uncertain whether it is the most probable reason. Caution should be exercised when making comparisons between data obtained from various instruments and missions.

Lines 123-124: The determination of whether there is a variation with time or not has not been firmly established. Despite this uncertainty, the assumption that Saturn's Bond albedo remains constant over time remains valid.

Lines 137-138: The authors mention that many current models and theories assume an energy budget in equilibrium over time. Providing some references to these models/theories would be beneficial.

Lines 139-149: The authors identify three primary factors driving time variability: large eccentricity, significant ring modulation, and giant convective storms occurring every 30 years. However, they do not provide specific quantitative contributions or indicate which factor is most determinant. Additionally, the presence or not of other potential factors is not addressed.

Lines 152-153 or 171: references to previous internal heat values in the literature? Line 171 provides one reference.

Lines 178-179: It is unspecified what is "our new investigation" and what "the previous study" is. References to both studies would enhance clarity.

Minor suggestions:

Fig. 1: Please write Voyager instead of Voy.

Throughout the tex, also in the SI: I prefer the use of microns rather than nm for wavelength units, especially for CIRS data in the infrared domain.

Fig. 4: The scale limits of Power per unit layer are different for the three plots. Using similar values would facilitate a more direct visual comparison.

Supplementary Information:

Lines 82, 111: Did the authors also attempt a period of 29.5 and compare both results?

Reference: Blake et al. 2023. Icarus 392.

Fig. S5: The scale limits of Emitted Power per unit layer are different for the six plots. Using similar values would facilitate a more direct visual comparison.

Lines 147-172: Similar to the rings, does Enceladus' water plume also impact the radiant energy budget, and if so, to what extent? Considering the potential effect of the Enceladus' water plume in the discussion, with references to Hartogh et al. (2011, A&A Volume 532) and Villanueva et al. (2023, Nature Astronomy Volume 7), would be beneficial.

Line 220: "Northern Hemisphere" is already defined with the acronym NH earlier, it can be removed here.

Lines 259-260: please specify where in the SI will be used the hemispheric calculations.

Lines 282-283: acronyms of "Northern Hemisphere" and "Southern Hemisphere" can be given here.

Line 326: please indicate the used CISSCAL release or version number.

Lines 346, 364, 371, 373, 394, 397: One aspect to consider in a manuscript is the reproducibility of the results. The authors mention images, and providing specifications or identification of these images in the manuscript, such as observation names found in the archive, or as a table with relevant parameters (e.g. obsid, start time, date, etc.), may enhance clarity and reproducibility.

Line 347: please indicate the used software version number.

Line 436: please indicate the used software version number.

Figures S16 and S17, illustrating reflectance measurements at three distinct times and various phase angles (62 and 39 degrees, respectively), lack sufficiently compelling evidence to substantiate the authors' assertion of variability. The relatively large error bars in the results do not conclusively establish variability. Including additional data points with reduced error bars could enhance the robustness of the study and further contribute to its advancement.

Line 498: please specify ESO telescope and instrument.

Lines 513-518: instruments? Resolutions (when not indicated)?

Lines 530-534: The authors justify some sources of the differences between the ESO and Cassini spectra. It is also noteworthy that the ESO spectrum of Saturn was acquired without rings.

Lines 536-537: Please specify the ESO instrument.

Lines 710-711, 715-716: see comment about figures S16 and S17. Rephrase the sentence accordingly.

Lines 750-752: specifications of the observations? See comment above regarding lines 346, 364, 371, 373, 394, 397.

Line 785: lines 772-774 indicate that the authors assume that the Bond albedo does not significantly change between the two hemispheres. Rephrase the sentence which mentions <“probably” does not vary>.

Reviewer #1 (Remarks to the Author):

This is a very interesting paper and represents an impressive amount of work in analysing a huge quantity of data. New estimates are presented of the Bond Albedo of Saturn and new insights gained into the radiative balance in this planetary atmosphere.

Although the work is substantive and important, I found it rather difficult to follow. Because of the format of Nature publications, the main paper has to be concise with details of the analysis presented instead in the Methods section, or in supplementary information. In this paper, I felt that this division had gone perhaps too far, with the main paper reduced almost to an extended abstract, and most of the scientific details of the paper contained substantially in the supplementary information paper. As a result, in order to make sense of what the authors have done the reader has to read both papers in parallel, which is difficult and time-consuming.

As to the work itself, it is mostly very impressive, but I felt there were a number of assumptions made that were not fully defended. That doesn't mean that I think these assumptions are wrong, it's just that I felt the assumptions needed to be better justified and evidence provided.

Reply: Firstly, we wish to extend our sincere gratitude to the reviewer for his or her diligent efforts in carefully reviewing our manuscript and providing numerous constructive suggestions. It is evident that the reviewer dedicated considerable time to this task, which we deeply appreciate. The manuscript has been notably enhanced as a result of addressing these valuable suggestions. We have addressed all of the suggestions and comments outlined below. Specifically, we have conducted additional analyses and provided further justifications and discussions regarding these assumptions used in the manuscript.

Furthermore, we concur with the reviewer's assessment that certain significant results and discussions should be integrated into the main text rather than being relegated to Supplementary Information (SI). In light of the guidelines provided by Nature Communications, allowing for up to 10 figures and 5000 words in the main text, we have restructured the manuscript accordingly. This involved incorporating more results and discussions from SI to the main text, along with maintaining adherence to the specified limits (10 figures and ~ 5000 words).

Additionally, as per the reviewer's recommendation, the remaining discussions initially included in the SI have been moved to the Methods section. Presently, the SI exclusively comprises supplementary figures.

Particular worries I have are:

1. The Cassini CIRS data cover only just over a 1/3 of one Saturn year, so most of the thermal component is estimated from an extrapolation, which is not, in my view, very well explained or justified.

Reply: Yes, we agree with the reviewer that the extrapolation from Cassini CIRS data needs more explanations and justifications. We collect more data, especially in the time after the Cassini epoch, to validate the extrapolation. A new study (Blake et al., 2023), based on the ground-based mid-infrared observations, analyzed the brightness temperature of Saturn. Specifically, this study provides the brightness temperature recorded at $17.7 \mu\text{m}$ from 2017 to 2022 (see Fig. 1), which is a period following the Cassini epoch. Additionally, the $17.7 \mu\text{m}$ observations sound the upper troposphere. Our previous study (Li et al., 2010) suggests that Saturn's emitted power mainly comes from the upper troposphere, so the $17.7 \mu\text{m}$ observations can be used to estimate Saturn's emitted power.

Figure 1. Ground-based observations of Saturn's brightness temperature from 2017 to 2022, which is a period following the Cassini epoch. The observations, which were recorded at a wavelength of $17.7 \mu\text{m}$, sound the upper troposphere of Saturn. The data come from a recent study by Blake et al. (2023).

The post-Cassini period from 2017 to 2022 corresponds to the summer season of the NH. The view geometry of the ground-based observatories, along with the effects of rings (e.g., blocking), make it challenging to observe the entire SH. Consequently, the ground-based observations (Blake et al., 2023) from 2017 to 2022 cover latitudes from $\sim 10^\circ\text{S}$ to 90°N , which do not resolve the whole SH. We first computed the NH-average brightness temperature to validate the extrapolation of NH-average emitted power. Subsequently, we used the average over limited latitudes in the SH ($0\text{-}10^\circ\text{S}$) to represent the SH-average brightness temperature. This was then

combined with the NH-average brightness temperature to estimate the global-average emitted power.

While brightness temperature is related to effective temperature (i.e., emitted power), they are not equal (Li et al., 2010). Our focus here is on examining the temporal variations of our extrapolation of emitted power, so we utilize the temporal variations of brightness temperature to estimate the temporal variations of effective temperature (i.e., emitted power). We use NH and global averages of brightness temperature from 2017 to 2022 to represent effective temperature and then compute the corresponding emitted power. Since brightness temperature is not equal to effective temperature (Li et al., 2010), the computed emitted powers from 2017 to 2022 are scaled to the finally estimated emitted power using the ratio between the emitted power computed from brightness temperature and the Cassini-CIRS measured emitted power in 2017. In other words, we shift the computed emitted powers based on ground-based observed brightness temperature (2017-2022) to align the computed 2017 emitted power with the corresponding CIRS-measured emitted power. Subsequently, the finally estimated emitted power at the global and NH scales are compared to the extrapolations, as shown in panels C and F of Fig. 2 (i.e., Fig. 4 of the main text in the revised manuscript). The temporal variations of the estimated emitted power from brightness temperature are generally consistent with our extrapolated emitted power based on the Cassini/CIRS measurements. This suggests that observations after the Cassini epoch qualitatively validate the extrapolation of Saturn’s emitted power.

Figure 2. Fitting Saturn’s global-average and NH-average emitted powers and extrapolating them from the Cassini epoch to the entire orbital period. (A) Modified global-average emitted power. The original emitted power was modified by removing the effect of the 2010 giant storm (see discussion in the supplementary Information). (B) Fitting the modified emitted power by a sine function with a fixed period of one Saturn year (29.4 years). (C) Extrapolating the global emitted power from the Cassini epoch (2004-2017) to the entire orbital period (1995-2025). The fitting results shown in panel B are used for the extrapolation. The results in (D), (E), and (F) are the same as the global analyses in (A), (B), and (C) respectively except for the NH. Vertical lines in panels C and F represent uncertainties. The global-average emitted power before the Cassini epoch (i.e., “validation 2” with the green dot in panel C) comes from a previous study¹⁸. The emitted powers after the Cassini epoch (i.e., “validation 1” with magenta dots in panel C and “validation” with magenta dots in panel F) are based on the ground-based observations of Saturn’s brightness temperature from a recent study (Blake et al., 2023).

A previous investigation of Saturn’s global-average emitted power (Erickson et al., 1978) is based on relatively high-quality observations conducted in 1971-72. Unfortunately, the observational times are beyond the whole orbital period including the Cassini epoch (1995-2025). We add the observational time (1971-72) by a one orbital period (~ 29.4 years) to project the previous study to year 2001 and then compare with our extrapolation of the global-average emitted power (the green point in panel C in Fig.4). The comparison shows that the difference between the previous result (Erickson et al., 1978) and the extrapolated value is less than the uncertainty of the extrapolated value, which suggests that they are qualitatively consistent.

Estimating the uncertainties of the extrapolation from the Cassini epoch to the entire orbital period is challenging. As we mentioned in the original manuscript, our analysis of the uncertainties probably underestimates them especially for these years far away from the Cassini epoch. Here, we provide a new and better estimate on the uncertainties related to the extrapolation. We combine the uncertainty in the measurements of the emitted power based on the Cassini observations with the fitting residuals (i.e., fitting results minus observational data) to estimate the uncertainties of the extrapolated emitted power for the years neighboring to the Cassini epoch. For example, the uncertainty of the 2017 measurement of Saturn’s emitted power is combined with the sine-function fitting residual for the 2017 observational data to estimate the uncertainty of the extrapolated emitted power in 2018. Likewise, we can combine the measurement uncertainty and the sine-function fitting residual at the 2005 observational data to estimate the uncertainty of the extrapolated emitted power in 2004.

The above estimates probably work for these extrapolated emitted powers at times neighboring the Cassini epoch (2004 and 2018, respectively), but they may underestimate the uncertainties for the extrapolated emitted powers with times far away from the Cassini epoch. Here, we use the standard deviation of the measured emitted powers during the Cassini epoch, which is much larger than the above-mentioned uncertainties, as the upper limit of the uncertainties for the extrapolated emitted power. Given their distance from the Cassini epoch, the initial and final years of the entire orbital period (1995 and 2025, respectively) are expected to bear the greatest uncertainty, corresponding to the standard deviation during the Cassini epoch. Subsequently, we linearly interpolate the uncertainty of emitted power at the beginning of the Cassini epoch (2005) and the largest uncertainty in the initial year of the whole orbital period (1995) to get the uncertainties for the extrapolated emptied powers from 1995 to 2004. Likewise, we linearly

interpolate the uncertainty of emitted power at the end of the Cassini epoch (2017) and the largest uncertainties at the final year of the whole orbital period (2025) to get the uncertainties for the extrapolated emitted powers from 2018 to 2025 (see Fig. 2).

It should be mentioned the large uncertainties in the extrapolation of emitted power do not significantly affect the conclusion in the manuscript. The energy imbalances suggested from this study for both Cassini epoch and the whole orbital period are mainly caused by the different seasonal variations between the absorbed power and the emitted power. Figures 9 and 10 in the revised manuscript show that the seasonal variations are much stronger in the absorbed power than in the emitted power. During the Cassini epoch, the emitted power changed less than 4% but the absorbed power changed more than 30%. Unless the emitted power in the periods beyond the Cassini epoch goes extremely away from the Cassini values, the time-average emitted power over the whole orbital period will be close to the time-average emitted power over the Cassini epoch. But it is basically impossible that the emitted power varies extremely beyond the Cassini epoch because of the 2010 giant storm, which is one of the strongest storms, only modified the global-average emitted power by ~2% (Li et al., 2015). Therefore, we do not think that the extrapolation of emitted power will change our conclusion – the seasonal energy imbalance on Saturn.

The extrapolations of Saturn's emitted power from the Cassini epoch to the entire orbital period, the new analysis of their uncertainties, and the validations of the extrapolations are all put in the main text, as the reviewer suggested (see Figs. 2-4 and lines 135-157 and lines 173-215 in the revised manuscript).

2. A very different estimate of the Bond Albedo is derived from the Cassini data to that resulting after the Voyager flybys. The authors suggest a reason why this might be different, but could, I think, have done more in demonstrating their hypothesis to be true. To be specific, the authors state that the Voyager estimate comes from analysing latitudes 11S to 32S, which is a dark latitude band, while their value is a true global average. Why couldn't the authors have demonstrated that if they also only considered 11S to 32S in their data that they'd get the same answer as the previous estimate?

Reply: We agree with the reviewer that the difference of Bond albedo between the current study based on the Cassini observations (2004-2017) and the previous study (Hanel et al., 1983) based on Voyager observations (1980-1981) should be investigated further. First, we examine the possibility of temporal variation of Saturn's Bond albedo from the Voyager time to the Cassini epoch. The Cassini analysis (Figs. S13 and S14) reveals that Saturn's full-disk reflectance and Bond albedo did not significantly vary during the Cassini epoch. Additionally, Earth-based observations (Karkoschka et al., 1995, 1998) also suggests that the Saturn's full-disk reflectance and hence Bond albedo basically remained constant from 1993 to 1995. Finally, a previous study conducted by one of the co-authors of the current manuscript (Mallama, 2012) also indicates that that there was no significant difference of Saturn's full-disk reflectance between the observations during the 1963-1965 period (Irvine et al., 1968a, 1968b) and those during the 1991-2009 period (Schmude, 2011). These studies suggest that Saturn's full-disk Bond albedo did not significantly change from the Voyager epoch (1980-1981) to the Cassini time (2004-2017). Therefore, we think that the difference of Bond albedo between the Voyager and Cassini measurements are caused by somethings else.

We have also followed the reviewer's suggestion to compare the reflectivity between the belt from 11S to 32S and the global average. We find the reflectance decreased by 4.6% from the

global-average reflectivity to the belt reflectance. The the difference of Bond albedo between the Cassini measurements (0.41+/-0.02) and the Voyage analysis (0.34+/-0.03) has a lower limit (0.37-0.39)/0.37 ~ -5.4% and an upper limit (0.31-0.43)/0.31 ~ 38.7%. The difference of reflectance between the global average and the result of the 11-32°S belt is close to the lower limit of the Bond-albedo difference between the Cassini and Voyager measurements, but it cannot explain all the difference.

Reproducing Voyager’s analysis of Saturn’s Bond albedo and further discerning the differences between Voyager’s analysis and the Cassini measurements is challenging due to the lack of necessary details analyzing the Voyager data in the previous study (Hanel et al., 1983) and the difficulty of reprocessing the old Voyager datasets. It should be mentioned that our study is based on long-term multi-instrument Cassini observations, which exhibit significant improvements in various aspects (such as much better coverage of latitude, wavelength, and phase angle) compared to the Voyager observations used by the previous study (Hanel et al., 1983). Therefore, we think that the Bond albedo generated by our study is more robust.

We have added above discussions to the main text of the revised manuscript (see lines 283-312).

3. I have a minor concern with the fitted phase functions to some Cassini/ISS wavelengths, which I detail later.

Page 22, Figure S23. How realistic are the forward-scattering peaks in this plot, especially in panels F and G? If the fitting model was physically based then we might be happy, but as it's just a 4th-order polynomial there's no physical basis for this as far as I can see.

Page 22, line 563. ‘polynomial function works well’ Well it obviously fits the data well, but how trustworthy is the extrapolation from a simple 4th-order polynomial like this?

Page 22, line 568. The data sort of show the forward scattering properties described. However, the fitted curves are not, in my view, demonstrated to be reliable.

Reply: As the reviewer pointed out, this concern is detailed with other minor comments regarding the polynomial-function fitting of Saturn’s full-disk reflectance. Therefore, we put the concern and the comments together and address here.

We conducted more analysis by following the reviewer’s constructive suggestions. Yes, we agree with the reviewer that the data fitting needs more justifications including validating the forward scattering. We try physically-based functions to constrain the light-curve behaviors especially for the forward scattering at high phase angles. The double Henyey-Greenstein (H-G) function $P(A_{HG}, g_1, g_2, f, \phi)$ (Henyey & Greenstein 1941; Hapke, 2002), which is defined as below can resolve both forward and backward scattering. The double H-G function is expressed as below.

$$P(A_{HG}, g_1, g_2, f, \phi) = A_{HG} \cdot (f P_{HG}(g_1, \phi) + (1-f) P_{HG}(g_2, \phi))$$

where A_{HG} is the coefficient to match the amplitude of the observed phase function. The term $P_{HG}(g, \phi)$ represents both forward (with a factor g_1 and $g_1 \in [0,1]$) and backward (with a factor g_2 and $g_2 \in [-1,0]$) scattering lobes, respectively. The factor f ($f \in [0,1]$) stands for the

fraction of the forward versus backward scattering. The term $P_{HG}(g, \phi)$ (g can be g_1 or g_2 and ϕ is phase angle) has a form as $P_{HG}(g, \phi) = (1 - g^2) / (1 + g^2 + 2g \cdot \cos \phi)^{3/2}$. The fitting by the double H-G function is compared with the polynomial-function fitting in the following figure.

Figure 3. Examples of fitting the phase function of Saturn's full-disk reflectance. (A) Fitting results for the combined data between the ESO observations and the Cassini ISS observations recorded at the RED filter (~ 647 nm). A four-order polynomial function (green line) and the double H-G function (black line) are used for fitting the data. (C) Fitting residuals (i.e., fitting results minus observational data) for the fittings shown in (A). Fitting residual 1 is for the fitting with a four-order polynomial function and fitting residual 2 is for the fitting with the double H-G function. Panels (B) and (D) are the same as panels (A) and (C) respectively except for the combined data between the ESO observations and the Cassini ISS observations recorded at the strongest methane-absorption filter (i.e., MT3 ~ 890 nm).

Panels A and B show that Saturn's full-disk reflectance displays stronger forward-scattering in the methane-absorption filters (e.g., MT3) (panel B) than in other filters (e.g., RED)

(panel A). Specifically, the MT3 data (panel B) shows that the reflectance is higher for the phase angles around 150° than around 130° , which demonstrates the relatively stronger forward-scattering at the methane-absorption wavelengths. Additionally, both the polynomial-function and double H-G fittings capture the forward-scattering property (panel B). However, the comparison of fitting residuals (panels C and D) suggests that the polynomial-function fitting is better than the double H-G fitting with relatively smaller fitting residuals. The physically-based functions can shed light on the atmospheric properties of giant planets (Heng and Li, 2021), but here we seek to fit the phase function of full-disk reflectance as good as possible to provide a relatively precise measurement of Saturn's Bond albedo. Therefore, the polynomial-function fitting is applied in our analysis. It should be mentioned that the fitting at these high phase angles lacking observations (e.g., $\sim 150\text{-}180^\circ$) may have relatively large uncertainty. But the reflectance at the relatively high phase angles does not significantly contribute to the Bond albedo because there is a factor of sine of phase angle in computing Bond albedo (Conrath et al., 1989; Hanel et al., 2003; Li et al., 2018, 2023) and this factor is small when phase angles approaching to 180 degrees. Therefore, the fitting uncertainties at the high phase angles, which are considered in our estimate of total uncertainty, do not significantly affect our analysis of Saturn's Bond albedo.

The above figure (Fig. 3) has been added to the main text of revised manuscript (Fig. 6) and the corresponding discussion is also put in the main text (Please see lines 247-256 and lines 264-270 of revised manuscript).

My remaining comments are mostly minor and are listed below for the main paper and the supplemental information paper.

Main article

Abstract: Many 'probables' and 'possibles' are listed in this abstract. Is this perhaps a bit too tentative? I think maybe it would be better to mention those things that the authors are confident of with a bit more certainty.

Reply: We have followed this suggestion to remove some 'probables' and 'possibles'.

Page 3, line 59: delete 'extremely'?

Reply: We have followed this suggestion.

Page 3, line 68: delete 'of'

Reply: We have followed it.

Page 3, line 69: Sentence 'Such investigation is further used to re-examine the internal heat.' This sentence reads awkwardly and could be reworded.

Reply: We have reworded it to "The investigation of the seasonal variations of Saturn's radiant energy budget is further used to estimate the internal heat".

Page 4, line 82: The Cassini CIRS data cover just 38% of the Saturn year and so much of these results depend on how these data were extrapolated to cover the rest of the year. But the details of this crucial extrapolation are buried within the extensive

supplementary paper. I think details of this extrapolation should be included in the main paper.

Reply: We agree with the reviewer and we have moved the discussion of extrapolation with suggested investigations (e.g., validations with other data/studies) to the main text (please see our reply to your first major concern).

Page 4, lines 88 – 93. An awful lot of the details of this study are in the supplemental material, which makes this really two papers in parallel, and thus rather hard to follow.

Reply: We have done two things to follow this great suggestion: (1) We have moved some important discussions from the SI to the main text; and (2) the remaining discussions in the SI are merged into the Methods section.

Page 4, line 95. The ‘significant increase in emitted power in the middle latitudes of the NH from 2010 to 2011’ is hard to see (in my opinion) from Figure 1. A time series of the emitted power at NH mid-latitudes would show this better. A good figure to show this is Fig. S3 in the supplemental information.

Reply: Yes, Fig. S3 is moved to the main text (see Fig. 2 of revised manuscript).

Page 5, line 109. Is it possible that the ‘significant’ difference between the Voyager and Cassini observations are due to systematic offsets between Cassini and Voyager, rather than ‘climate change’? How do we know that the two datasets are directly comparable? I think some justification is needed.

Reply: We agree with the reviewer that we cannot confidently conclude that the difference between Voyager and Cassini observations is due to climate change. Therefore, we reword the sentence to “The difference between the two observations provides an opportunity to examine the possible interannual variations of Saturn’s emitted power.” (see lines 113-114).

Page 5, lines 118-121. The authors say that the difference between their derived Bond albedo than that deriving from Voyager observations may arise from the Voyager value coming only from observations from 11S to 32S. Couldn’t this explanation be verified immediately by the authors reanalysing their Cassini data using the 11-32S latitudes only? Then, the authors would be able to compare directly with the Voyager estimate and presumably get the same answer, assuming this hypothesis is correct.

Reply: We have followed the reviewer’s suggestion to do more analysis and add the corresponding discussion into the main text (please see lines 283-312 of the revised manuscript). Please also refer to our reply to your second major concern.

Page 6, line 125. Change ‘changes’ to ‘change’

Reply: We have followed it.

Page 6, line 133. The fact that Saturn emits more than it absorbs from the Sun was already very well known from Voyager times. Perhaps add a note to this effect here?

Reply: We have followed it by adding the corresponding references (please see lines 323-326).

Page 6, lines 133-135. The use the word 'deficit' and then negative values and words

such as maximum and minimum is confusing. The deficit, to me, implies a negative value. Hence, I would write this as, 'the radiant energy deficit (i.e., outgoing - incoming) changed from a minimum value of 2.63 W/m² in 2003 to a maximum value of 3.05 W/m² in 2013.

Reply: We have basically followed this suggestion by removing “maximum” and “minimum”. We believe it is better to retain the minus sign before the numbers 2.63 and 3.05, as both numbers represent deficits. We reword the sentences to “Therefore, there is a significant radiant energy deficit, which is defined the difference between the absorbed solar energy and the emitted thermal energy (i.e., the absorbed solar energy minus the emitted thermal energy), for Saturn as a whole (including its atmosphere and interior)^{12,17-19}. This energy deficit suggests that Saturn is losing energy, a phenomenon referred to as global cooling. The range of the energy deficit spans from -2.63±0.08 Wm⁻² in 2003 to -3.05±0.07 Wm⁻² in 2013.” (please see lines 323-328).

Page 6, line 137. Which ‘current models and theories’ are these? Add some references here, please.

Reply: We have followed this good suggestion by adding some references (please see lines 330-332).

Page 6, line 148. The emitted power curve is mostly extrapolated from the CIRS observations. In fact, the CIRS observations account for only 38% of the whole Saturn year. Hence, a lot of this paper's conclusions depend on the reliability of this extrapolation. I think this should be discussed more.

Reply: We follow the reviewer’s suggestion to do more analysis and add the corresponding discussion into the main text (please see lines 173-215 of the revised manuscript). Please also refer to our reply to your second major concern. It should be mentioned the large uncertainties in the extrapolation of emitted power do not significantly affect the conclusion in the manuscript (i.e., seasonal energy imbalance on Saturn), as we reply to your second major concern.

Page 7, line 155. ‘explored in previous studies’. Again, can you add references please?

Reply: We have followed this good suggestion by adding some references (please see line 350).

Page 7, line 156. ‘led to a significant issue’. Can you be more precise - what 'significant issue' is this?

Reply: We reworded the sentences to “This variation must be considered when estimating the seasonally constant internal heat. Such seasonal variations were not fully explored in previous studies of giant planets^{12,17-19}, leading to less precise estimates of their internal heat.”.

Page 7, line 158. ‘covered three of Saturn’s seasons’. I think this is a bit misleading. The observations in Fig 3C cover 2006 - 2017, i.e., 11 years. Compared with a Saturn year of 29.4 years. That's 38% of the year, which is hardly three seasons. It's not even two! I guess you could argue that you got all of southern spring, and small parts of the seasons either side, but that's not quite what is implied by this statement.

Reply: We have reworded the sentence to “The long-term Cassini observations partially covered three of Saturn’s seasons (i.e., part of the NH winter, complete NH spring, and part of NH summer), providing a great opportunity to estimate Saturn’s internal heat.” (please see lines 351-353).

Page 7, line 167. I'm not sure CIRS captured the dominant part of the cycle. You have extrapolated the outgoing radiation either side of the Cassini Epoch so it's hardly surprising that the two time-ranges give similar values.

Reply: We basically agree with this comment. As discussed in our previous studies (Li et al., 2010, 2015), Saturn's emitted power is mainly determined by the temperature of upper troposphere, which is further determined by the solar flux. Considering the solar flux has the dominant seasonal variation during the Cassini epoch, it is possible that Saturn's emitted power has the dominant seasonal variation during the Cassini epoch too. But it is possible that there are other factors affecting Saturn's emitted power. Therefore, we agree with the reviewer to that we cannot confidently get such a conclusion. We have reworded the sentence to "The seasonal cycle of solar constant is one of the key factors affecting the seasonal variations of Saturn's radiant energy budget^{15, 16}." (please see lines 365-367).

Page 7, line 168. 'This is further supported...' Aren't you saying the same thing again here?

Reply: We have reworded the sentences to "It helps explain the consistency of the estimated time-average internal heat between the complete orbital period ($2.84\pm 0.20 \text{ Wm}^{-2}$) and the Cassini epoch ($2.89\pm 0.18 \text{ Wm}^{-2}$)." (please see lines 367-369).

Page 10, line 248. This is not really a very great difference between the obliquities of Saturn and Neptune.

Reply: We have followed this suggestion to remove the comparison of obliquity between Saturn and Neptune. The sentence now reads, "However, Neptune has a significant obliquity ($\sim 28.3^\circ$), implying that it experiences a hemispheric-average seasonal energy imbalance caused by its obliquity". (please see lines 450-452).

Page 10, line 249. I don't agree. If Neptune's eccentricity is small then there shouldn't really be much difference between the balance in the northern and summer hemispheres.

Reply: Eccentricity mainly affects the seasonal variations of global-average solar flux and hence the seasonal variations in the global-average absorbed solar energy due to the varying distance between a planet and the Sun. On the other hand, obliquity affects hemispheric seasonal variations by differentiating solar flux to the two hemispheres. For example, Earth has a small orbital eccentricity but a significant obliquity. Therefore, Earth does not experience significant seasonal variations in global-average solar flux and absorbed solar energy, but it does have significant seasonal variations in hemispheric-average solar flux and absorbed solar energy. Generally, the seasonal variations of emitted power are much smaller than the seasonal variations of absorbed solar power for bodies with significant atmospheres (Jacobowitz et al., 1979; Yang et al., 1999; Li et al., 2015; Creedy et al., 2021). Therefore, Earth has a significant seasonal energy imbalance at the hemispheric scale. We think that Neptune exhibits similar characteristics to Earth.

We realize that we may not have made the above point clear in the original manuscript, and we have reworded the sentence to: "However, Neptune has a significant obliquity ($\sim 28.3^\circ$), implying that it experiences a hemispheric-average seasonal energy imbalance caused by its obliquity." (Please also refer to our response to your previous comment).

Methods. This is a very short methods section, mostly referring the reader to other papers. I would have thought much of the information now in the supplementary paper could go here.

Reply: We have followed this great suggestion to move the supplementary information to the method. Now the supplementary information only includes supplementary figures only.

Page 14, line 327. Fig. 1 Caption. Change 'are' to 'is'

Reply: We have followed it.

Page 14, line 331. Fig. 1 Caption. The 'dramatic increase of emitted power in the NH between 2010 and 2011' is hard to see on this plot as it is difficult to pick out the individual lines. It doesn't look any more significant than other inter-annual changes. Fig. S3 is better for this.

Reply: We have followed this suggestion to remove the statement from the caption of Fig. 1. Additionally, we have added Fig. S3 (now Fig. 2 in the revised manuscript) to the main text.

Page 15, line 342. Fig. 2 Caption. These data are from Cassini SSI and VIMS. I think the caption should state this explicitly.

Reply: We have added such a statement "Results are mainly based on the observations recorded by the Cassini spacecraft (ISS and VIMS) and ESO." in the caption.

Page 16, line 357. Fig. 3 caption. The black and magenta lines in Fig 3C are explained here, but I think it would be neater and easier to follow if these labels could be marked on the figure itself in some way.

Reply: We have followed this good suggestion. Please see the new figure (Fig. 9 in the revised manuscript).

Page 17, line 362. Fig. 3 caption. What is the evidence that the Bond Albedo is constant with seasons? This is discussed in the manuscript, but this statement in the caption should be qualified.

Reply: We have removed the statement from the caption. Instead, we explicitly state that the absorbed solar power comes from Fig.S25. Additionally, the discussion of temporal variations of Bond albedo has been moved to the main text. The original Fig. 3 is changed to Fig. 9 because we have added more figures into the main text by following your suggestion (see the new caption of Fig. 9).

Page 17, line 368. Fig. 3 caption. 'extrapolation to the entire orbital period are discussed in the first section of SI.' This matter of how the CIRS data were extrapolated to other seasons seems really central and important to me, so I'm puzzled that it's been relegated to the supplementary information.

Reply: We have followed this good suggestion by moving the discussion to the main text. Additionally, we have added more discussion on validating the extrapolation to the main text. Finally, we have replotted the figure by adding validations. Please refer lines 135-215 and the new figure (Fig. 4 in the revised manuscript).

Page 17, line 371. Fig. 3 caption. Briefly summarise how these data have been 'extrapolated'.

Reply: We have moved the extrapolation to the main text. Additionally, we have added discussion on the validations to the main text. Please see our reply to your previous comment.

Page 18, line 392. Add ', calculated as described in the text' after 'internal heat flux'.

Reply: We have followed it.

Page 18, line 394. Add '(extrapolated as described in the supplementary material)' after 'extrapolated results'.

Reply: Because we have moved the extrapolation from SI to the main text (see Figs. 3 and 4), we modify the section in the caption to "The output flux is determined solely by the emitted power. The global and hemispheric averages of emitted power come from Figs. 3 and 4."

Supplementary Article

Page 3, Fig. S3. Here, the sharp increase from 2009 to 2011 for the northern hemisphere is obvious. Please, refer to this figure from main text when you discuss this effect.

Reply: We have moved Fig. S3 to the main text (please see Fig. 2 in the revised manuscript).

Page 5, Fig. S5. The extrapolated curve in Panel B has a value of 4.85 in 1996, but the same curve has a value of 4.9 in 1996 in Panel C. How can this be?

Reply: Yes, that is a mistake. We have corrected the mistake (please see Fig. 4 in the revised manuscript).

Page 5, Figs S5A-C. How can a sine wave fitted to the modified data in Panel B match the original data in panel C? You must have modified the fit after 2017 in some way. The fact that these plots use different y-scales is not helpful. Something is missing here.

Reply: Yes, we agree with the reviewer that the original discussion is not clear. We have modified the discussion to "A previous study (Li et al., 2015) suggests that the 2010 giant storm increased the global-average emitted power by 2%. We first subtract such an increase (2%) from the observed global-average emitted power to get a modified emitted power for the period after the 2010 giant storm (i.e., 2011-2017), as shown in panel A of Fig. 4. We then fit the modified emitted power using a sine function with a period of 29.4 years. Panel B of Fig. 4 shows that this fitting works well. Finally, we use the fitting results to extrapolate the global-average emitted power from the Cassini epoch to the entire orbital period (panel C of Fig. 4). For the period after 210, we increase the modified emitted powers and the corresponding fitting results by 2% to revert the modified emitted powers from 2010 to 2017 back to the original measurements." (please see lines 159-168).

Page 6, lines 158-161. How can this be right? Saturn has generally stronger absorption bands in mid-IR, so more light would be absorbed before reflection. Also, the scattering

of cloud particles is less effective at long wavelengths. Hence, assuming the albedo is the same at thermal IR as it is in the near-IR is a BIG assumption. Is there no way of computing an effective albedo at longer wavelengths? What effect might errors in this assumption make?

Reply: Yes, this simple assumption probably overestimates the Bond albedo of the rings' thermal emission concentrating in the middle and long infrared wavelengths because the absorption and scattering of cloud particles are probably strong in these wavelengths. However, this assumption essentially does not affect our analysis of Saturn's radiant energy budget because the thermal emission from the rings is smaller by at least one order of magnitude compared to the scattering and blocking of the rings (Figs. S3 and S5). We have added the corresponding discussion into Methods (please see lines 543-549).

Page 6, line 175. Change 'on-orbit' to 'in-orbit'?

Reply: We have followed it.

Page 7, line 199. Diagram (Fig S6) covers 1996 to 2025, not 1995 – 2025.

Reply: The entire orbital period spans from November 1995 to May 2025. Since the start is near 1996, we utilize "1996 ... 2024" for the x-axis. However, in some plots, such as Figs. 9 and 10, we employ different x-tickers, i.e., "1995 ... 2025". Nevertheless, it is not ideal to leave blank areas for plotting 2-D contour figures, such as Fig. S6. Therefore, we use "1996 ... 2024" for Fig. S6.

Page 8, line 234. Change 'with considering' to 'including'.

Reply: We have followed it.

Page 11, Fig. S11. Can't you take the image A and roughly compute the reduction in solar irradiance caused by the ring shadow? You could assume N/S hemispheric asymmetry and then calculate how much the expected reflection has been reduced by the shadow as a function of wavelength and then compare directly with Panel D.

Reply: Panel D displays the outputted solar power per unit area from rings mode, expressed in units of "W/m²". Panel A shows the calibrated solar irradiance, measured in units of "photons/s/cm²/sr". While it is possible to convert solar irradiance to solar power using assumptions regarding the dependence of solar irradiance on wavelength and solid angle (sr), the challenge lies in estimating the original solar flux for areas shadowed by rings (i.e., the blue belts in the southern hemisphere of panel A). The reviewer suggests using the assumption of hemispheric symmetry to derive the original solar flux in the southern hemisphere, but such an assumption may not be valid. The sub-solar latitude of the ISS image shown in panel A is 9.5°N, and the latitudes shadowed by rings in the southern hemisphere are around 10°S (see panel C). In the northern hemisphere, the latitudes around 10°N (which are close to the sub-solar-latitude at 9.5°N) receive the highest solar irradiance and consequently reflect more solar irradiance, making them significantly greater than those in latitudes around 10°S in the southern hemisphere. Additionally, the sub-solar latitude (9.5°N) and the sub-Cassini latitude (0.2°N) result in significantly different view geometries and phase angles for the observed reflected solar irradiance between the latitudes around 10°N and the latitudes around 10°S, which also invalidate the assumption of hemispheric symmetry. Therefore, we cannot utilize solar irradiance data from the northern hemisphere to determine the original solar irradiance in latitude bands around 10°S in the southern hemisphere.

Page 12, line 307. Is it safe to assume that ‘the effects of ring-shadowing and ring-scattering do not vary significantly with wavelength’? What is the evidence that this assumption is sensible?

Reply: The optical depths of the rings are generally greater than 1 in the visible and near-infrared wavelengths (Colwell et al., 2009, 2010), indicating that ring-shadowing is independent of wavelength. This assertion is further supported by Cassini/ISS observations (Fig. S23). Additionally, ring-scattering does not vary significantly with wavelength in the visible and near-infrared spectra (Franklin et al., 1965; Nicholson et al., 2008), which primarily contribute to solar flux. We have added the above discussion to the Methods (see lines 617-621).

Page 12, line 323. Fig. S12 Caption. What ISS wavelength is this?

Page 13, line 358. Fig. S13 Caption. Again, what ISS wavelength is this?

Reply: We have followed the above two suggestions. In fact, we have provided not only the wavelength but also image identification number for all ISS and VIMS images in these supplementary figures to facilitate searching and reproduction by other researchers, as suggested by the other reviewer (please see captions of Figs. S8, S9, S10, S11, S13, S14, S15, S22, and S23).

Page 13, line 366. Change ‘built Saturn’s’ to ‘constructed’

Reply: We have followed it.

Page 19, line 498. Please give more details on what these ESO observations were, and which instrument/telescope they were made with. It’s not enough, I think, to just give a reference and expect the reader to figure this out for themselves.

Reply: We have provided the details such as instrument, telescope, and spectral range/resolution for EOS observations in the revised manuscript (please see lines 757-761).

Page 20, Figure S21. Can't these be combined into a single figure? It would make it easier to quantitatively compare the data. Also, in terms of energy balance, what really matters is the product of the reflectivity and incident solar irradiance. which will underline the dominance of the reflectivity at green wavelengths over all others. I suggest changing this figure to one with two panels - one showing the reflectivity from all three instruments in the same plot, and one showing the same data, but multiplied by the solar irradiance.

Reply: We have followed this valuable suggestion to put the three datasets together for ease of comparison, revealing consistency among them. However, we chose not to include the suggested second panel for reflected solar irradiance. Since the three datasets are based on observations from different years (and thus different seasons) (please see the caption of Fig. S17 in the Supplementary Information), the incident solar irradiance varies. If we were to multiply the consistent reflectance with the incoming solar irradiance, it would highlight differences in reflected solar irradiance among the three datasets, despite the reflectance remaining consistent. Therefore, we did not add such a panel.

Page 20, line 515. ‘European Southern Observatory (ESO)’ Again, what instrument is this? What telescope, in fact?!

Page 20, line 531. 'ESO observations'. Again, you need to define what instrument this was. And what telescope. It's presumably VLT but this should be stated here - the reader should not have to chase after references to learn this.

Reply: For the above two suggestions, please refer to my previous reply. We have provided the details such as instrument, telescope, and spectral range/resolution for EOS observations in the revised manuscript (please see lines 757-761).

Page 24, lines 607 – 619. I'm a little uneasy about this. As I understand it the authors have used the phase functions at CM3 and MT3 and applied these to longer wavelengths to extrapolate the VIMS observations. How much methane absorption was necessary to switch between the two phase functions? What about wavelengths where methane absorption is medium? Was it a simple binary switch between phase function depending on wavelength, or were the phase functions interpolated between in some way?

Reply: For wavelengths longer than 939 nm but shorter than the longest wavelength observed by VIMS (5131 nm), we use both the phase functions from the CB3 filter (939 nm) and MT3 filter (890 nm) to extrapolate the phase functions at wavelengths longer than 939 nm. The Cassini ISS observations and fitting results (Fig. S23) demonstrate that the phase functions differ between the continuum bands (e.g., CB2 and CB3) and the methane-absorption bands (e.g., MT2 and MT3), so we use the phase functions at the MT3 and CB3 wavelengths to scale these wavelengths with and without methane absorption, respectively. We use the methane-absorption band around 1400 nm (Fig. S16) as an example to demonstrate how to scale the phase angle at MT3 (890 nm) to obtain the phase function at the wavelength of 1400 nm. Firstly, we calculate the ratio of full-disk reflectance at a phase angle of $\sim 11.5^\circ$ between 1400 nm and 890 nm based on the spectra recorded by the Cassini/VIMS (Fig. S16). Then, we scale the complete phase function at 890 nm by this ratio to obtain the phase function at the wavelength of 1400 nm.

We have incorporated the above discussion into the Methods section (please refer to lines 840-849). It should be noted that solar irradiance beyond the wavelength of ISS/CB3 (939 nm) constitutes only a small fraction of the total solar flux. Furthermore, uncertainties associated with filling observational gaps in wavelength have been considered into our analysis. Therefore, the simple extrapolation of phase functions from the ISS CB3/MT3 wavelengths to wavelengths longer than 939 nm does not significantly impact the Bond albedo.

Additionally, we have included a figure (Fig. S20) to illustrate the filling of observational gaps in wavelength (please refer to lines 816-829 for the corresponding discussion).

Page 24, Fig. S25. What are these panels, exactly? Are these from VIMS data? Or do they also include the SSI and 'ESO' data? For the geometric albedo, what were the observing conditions of what I assume is a VIMS spectrum (i.e., what at the solar zenith angle, viewing zenith angle, and phase angle)? For Panel B, please define what a phase integral is. Is this panel attempting to show the transition between the CM3 and MT3 phase functions?

Reply: In the revised manuscript, we have removed two panels and retained only the panel for monochromatic Bond albedo (see Fig. S21). Monochromatic Bond albedo, defined as the Bond

albedo at each wavelength (Li et al., 2018, 2023), represents the ratio of absorbed solar irradiance to incoming solar irradiance at each wavelength (see lines 516-519). It directly influences the Bond albedo and absorbed solar power. Monochromatic Bond albedo is calculated by integrating full-disk reflectance (Fig. 8 in the main text) across phase angles at each wavelength, following the methodology described in our previous studies (Li et al., 2018, 2023).

Page 25, lines 631-632. As I noted earlier, can't you calculate the bond albedo from your data, limiting to these latitude bands, and verify that this accounts for the discrepancy between these estimates of the Bond Albedo?

Reply: Yes, we have conducted the suggested calculation and included further discussion on the difference in Bond albedo between the Voyager study and the current Cassini study (please refer to our response to your second major concern).

Page 26, lines 667-668. Can this 'cancelling out' of the noises at different wavelengths and phase angles be validated or estimated?

Reply: Yes, one calibration paper of the Cassini data (Knowles et al., 2020) suggests that the data calibration underestimates the reflected radiance at some wavelengths and phase angles but overestimates it at other wavelengths and phase angles. We have added the above sentence with the reference (please see lines 885-887)

Page 26, line 699. Why is this reference spelt out in terms of the authors, while all others are simply referred to by their numbers?

Reply: We have changed the sentences to “We follow the rule of error propagation of addition, as described in a previous study⁴⁶” to keep consistency.

Reviewer #2 (Remarks to the Author):

The authors investigated Saturn's radiant energy budget and its seasonal variability mainly by using the multi-instrument long-term observations from Cassini (spanning from 2004 to 2017, partially covering three of Saturn's seasons). Their findings reveal that the energy budget exhibits significant dynamical imbalances, seasonal variations of the planetary cooling, and higher Bond albedo and internal heat flux values than previous estimations.

These results are interesting, important and hold significance in the field, considering that the majority of current evolutionary and atmospheric models for giant planets operate under the assumption of a globally balanced energy budget across all time scales. This widely accepted assumption within the scientific community is primarily attributed to the lack of appropriate long-term observations, and it facilitates the development of complex models. Additionally, the radiant energy budget holds importance for internal heat estimations.

The authors conducted a careful analysis of archival Cassini/CIRS, ISS, and VIMS data, presenting it in a relative sufficient detail. They employed appropriate methods for

estimating both emitted and absorbed powers, as described in previous studies. As an additional aspect beyond the state-of-the-art, the authors introduced the computations of the effects of the Saturn rings on Bond albedo estimation.

Reply: We appreciate the words of encouragement and the positive comments provided by the reviewer. We sincerely thank the reviewer for his or her constructive suggestions, which have greatly contributed to improving the manuscript. By implementing the suggested improvements, the quality of the manuscript has been significantly enhanced.

Suggested improvements/comments:

Line 33: Similar to Jupiter and Uranus, it is expected that Saturn's overall energy budget is not in a stable state, which is unsurprising. Various factors may influence this equilibrium, such as the direct absorption of incoming solar irradiance by the atmosphere, limited thermal radiation from deeper levels, and the presence of a heat source. Please rephrase the sentence accordingly.

Reply: We have followed the reviewer's suggestion to remove the word "Surprisingly" (please see lines 33-34).

Line 55-57: Please add references to the sentence.

Reply: We have added a reference for this sentence (please see lines 55-58).

Lines 83-84: Please see comment in the SI below regarding lines 147-172.

Reply: Please see our reply to your comment on lines 147-172 below.

Lines 107-109: Temporal variability is a plausible factor, though it remains uncertain whether it is the most probable reason. Caution should be exercised when making comparisons between data obtained from various instruments and missions.

Reply: Yes, we have reworded the two sentences by following the reviewer's suggestion. The revised sentences now state, 'The Voyager profile differs from all Cassini profiles. Specifically, the solar longitude for the Cassini observations in 2010 is $\sim 10.1^\circ$, which is close to the solar longitude of $\sim 13^\circ$ for the Voyager observations in 1980-81. This suggests that the two observations are separated by \sim one Saturn year. The difference between the two observations provides an opportunity to examine the possible interannual variations of Saturn's emitted power' (please see lines 112-114).

Lines 123-124: The determination of whether there is a variation with time or not has not been firmly established. Despite this uncertainty, the assumption that Saturn's Bond albedo remains constant over time remains valid.

Reply: We have added more discussion on possible temporal variations of Saturn's full-disk reflectance and Bond albedo not only for the Cassini epoch but also for the longer time scale (please see lines 283-312).

Lines 137-138: The authors mention that many current models and theories assume an

energy budget in equilibrium over time. Providing some references to these models/theories would be beneficial.

Reply: We have added the references (please see lines 330-332).

Lines 139-149: The authors identify three primary factors driving time variability: large eccentricity, significant ring modulation, and giant convective storms occurring every 30 years. However, they do not provide specific quantitative contributions or indicate which factor is most determinant. Additionally, the presence or not of other potential factors is not addressed.

Reply: We have included quantitative contribution of three factors as “The solar constant at Saturn changes by $\sim 24.3\%$ from aphelion to perihelion. Secondly, the rings create a large modulation in the seasonal variations of the global-average solar flux with a magnitude $\sim 10.7\%$ (panel A of Fig. 9), which consequently impact the absorbed solar power (panel B of Fig. 9). Lastly, the occurrence of Saturn’s giant convective storms, approximately every thirty years^{61,62}, modify the emitted power and the absorbed solar power of Saturn. A previous study²⁵ and Fig. S14 suggest that the 2010 giant storm changed the global-average emitted power and absorbed power by $\sim 2.0\%$ and $\sim 2.9\%$, respectively”(please see lines 336-345).

Lines 152-153 or 171: references to previous internal heat values in the literature? Line 171 provides one reference.

Reply: We have added the references for this sentence (please see lines 349-350).

Lines 178-179: It is unspecified what is “our new investigation” and what “the previous study” is. References to both studies would enhance clarity.

Reply: We have revised the sentence to “These values are more compatible with the results of this study (Bond albedo 0.41 ± 0.02 and internal heat flux $2.84 \pm 0.20 \text{ Wm}^{-2}$) than the estimates (Bond albedo 0.34 ± 0.03 and internal heat flux $2.01 \pm 0.14 \text{ Wm}^{-2}$ from the previous study¹²)” (please see lines 377-379).

Minor suggestions:

Fig. 1: Please write Voyager instead of Voy.

Throughout the text, also in the SI: I prefer the use of microns rather than nm for wavelength units, especially for CIRS data in the infrared domain.

Reply: We first removed ‘Voy’ from the legend of the lines, keeping only ‘1980-81’ to maintain consistency with the legend of Cassini lines (‘2004-05, 2006, ...’). In the figure caption, we provide an explanation for the ‘1980-81’ line. For the ISS data, we use ‘nm’ instead of ‘micron’ because this unit was used in the Cassini ISS introductory paper (Porco et al., 2004) and many other ISS-related papers. Both ISS and VIMS were utilized to study the Bond albedo of Saturn. To maintain consistency, we also use the unit ‘nm’ for VIMS. However, we agree with the reviewer that the CIRS data in the infrared domain should use ‘micron’. The original unit of CIRS raw data is wavenumber (cm^{-1}), but we converted it to ‘micron’ (please refer to lines 493-495).

Fig. 4: The scale limits of Power per unit layer are different for the three plots. Using similar values would facilitate a more direct visual comparison.

Reply: Yes, we have basically followed this suggestion (please see Fig. 10 in the revised manuscript). The solar irradiance and the related absorbed power of the two hemispheres have opposite phases, which partially cancels out the seasonal variations at the global scale. This makes the magnitude of seasonal variations is smaller in the global analysis than in the hemispheric analysis. Given the varying ranges between global and hemispheric analyses, we employ different scales for global and hemispheric plots. However, we agree with the reviewer's suggestion that maintaining the same scale for the two hemispheric analyses would facilitate comparison (see panels B and C in Fig. 10).

Supplementary Information:

Lines 82, 111: Did the authors also attempt a period of 29.5 and compare both results?
Reference: Blake et al. 2023. Icarus 392.

Reply: The sidereal orbital period and tropical orbital period are 29.457 years and 29.424 years, respectively (see <https://nssdc.gsfc.nasa.gov/planetary/factsheet/saturnfact.html>). The tropical orbital period is defined as the time it takes for the Sun to pass from vernal equinox to vernal equinox, which effectively helps address the seasonal variations of planets. Therefore, we use the tropical year of Saturn (29.424 years) in this study. It is worth mentioning that there is a very small difference between the tropical and sidereal years (29.424 years vs. 29.457 years), and this difference basically does not affect the results. Additionally, the study by Blake et al. (2023) is referenced in this text, and the brightness temperature data provided in the study are used to validate the extrapolation of Saturn's emitted power (please refer to panels C and F in Fig. 4 and our response to the first concern from the other reviewer).

Fig. S5: The scale limits of Emitted Power per unit layer are different for the six plots. Using similar values would facilitate a more direct visual comparison.

Reply: We have moved Fig. S5 to the main text (Fig. 4) by following the suggestion from the other reviewer. As we have replied to your other suggestion, The solar irradiance and the related absorbed power of the two hemispheres have opposite phases, which partially cancels out the seasonal variations at the global scale. This makes the magnitude of seasonal variations is smaller in the global analysis than in the hemispheric analysis. Given the varying ranges between global and hemispheric analyses, we employ different scales for global and hemispheric plots. However, we agree with the reviewer's suggestion that maintaining the same scale for the hemispheric analyses would facilitate comparison, so we use a consistent scale for these panels of global analyses (see panels A-C in Fig. 4) and one more consistent scale for these panels of hemispheric analyses (see panels D-F in Fig. 4).

Lines 147-172: Similar to the rings, does Enceladus' water plume also impact the radiant energy budget, and if so, to what extent? Considering the potential effect of the Enceladus' water plume in the discussion, with references to Hartogh et al. (2011, A&A Volume 532) and Villanueva et al. (2023, Nature Astronomy Volume 7), would be beneficial.

Reply: Yes, we agree with the reviewer that the Enceladus water torus can potentially affect the radiant energy budget. However, its effects are much smaller than those from the rings due to its significantly colder temperature, greater distance from Saturn, and lower density compared to the rings. Therefore, the water torus generated by Enceladus is not considered in our analysis of Saturn's radiant energy budget. We have included the above discussion in the main text and referenced the two studies (Hartogh et al., 2011 and Villanueva et al., 2023) (please see lines 94-98).

The radiation area is comparable between the rings ($\pi \cdot 140,000^2 - \pi \cdot 70,000^2 \sim 4.6 \times 10^{10} \text{ km}^2$, where 70,000-140,000 km is distance range of rings refer to center of Saturn) and the Enceladus water torus ($2 \cdot \pi \cdot 230,000 \cdot 50,000 \sim 7.2 \times 10^{10} \text{ km}^2$, where 230,000 km and 50,000 km are the distance of the water torus from Saturn and the height of the water torus, respectively). However, the temperatures differ significantly between the rings (e.g., inner rings $\sim 110 \text{ K}$) and the Enceladus water torus ($\sim 25 \text{ K}$). Additionally, the distance from Saturn is largely different (70,000 km for the inner rings and $\sim 230,000 \text{ km}$ for the Enceladus water torus). Finally, the density of molecules is vastly different between the rings and the Enceladus water torus. Thermal radiation is proportional to the fourth power of temperature and inversely proportional to distance squared, so the differences in temperature and distance between the rings and Enceladus water torus will make the thermal radiation from the rings larger than that from the Enceladus water torus by at least three orders of magnitude. The density difference between the rings and Enceladus water torus will further magnify the difference of thermal radiation between them.

The thermal radiation from the rings is the smallest effect among the three effects from rings, at least one order of magnitude smaller than the other two effects (ring shadowing and ring scattering) (see Fig. S3 in the revised manuscript). Therefore, we think that the thermal radiation from the Enceladus water torus are much smaller than the effects of the rings and can be neglected.

Line 220: "Northern Hemisphere" is already defined with the acronym NH earlier, it can be removed here.

Reply: We have followed this suggestion.

Lines 259-260: please specify where in the SI will be used the hemispheric calculations.

Reply: It will be used in Fig. S25. We have followed this suggestion to specify.

Lines 282-283: acronyms of "Northern Hemisphere" and "Southern Hemisphere" can be given here.

Reply: We have followed this suggestion.

Line 326: please indicate the used CISSCAL release or version number.

Reply: We have followed this suggestion to add the version number (4.3) (please see line 655).

Lines 346, 364, 371, 373, 394, 397: One aspect to consider in a manuscript is the reproducibility of the results. The authors mention images, and providing specifications or identification of these images in the manuscript, such as observation names found in the archive, or as a table with relevant parameters (e.g. obsid, start time, date, etc.), may enhance clarity and reproducibility.

Reply: We have followed this good suggestion. In fact, we have provided the image identification numbers for all ISS and VIMS images, which are currently used by the PDS, to facilitate searching and reproduction by other researchers (please see captions of Figs. S8, S9, S10, S11, S13, S14, S15, S22, and S23).

Line 347: please indicate the used software version number.

Line 436: please indicate the used software version number.

Reply: We have provided the version numbers of the calibration software for ISS (CISSCAL 4.3) and VIMS (ISIS3) in Methods (see lines 655 and 714). Additionally, the version numbers are provided in the captions of Fig. S9 (ISS calibration) and Fig. S15 (VIMS calibration).

Figures S16 and S17, illustrating reflectance measurements at three distinct times and various phase angles (62 and 39 degrees, respectively), lack sufficiently compelling evidence to substantiate the authors' assertion of variability. The relatively large error bars in the results do not conclusively establish variability. Including additional data points with reduced error bars could enhance the robustness of the study and further contribute to its advancement.

Reply: We have searched the entire ISS dataset on the PDS. Unfortunately, the images depicted in Figs. S16 and S17 are all global images captured at the same phase angle. Nevertheless, we have included additional discussion in the main text regarding the potential temporal variations of Saturn's full-disk reflectance (see lines 283-294).

Line 498: please specify ESO telescope and instrument.

Reply: We have provided the details such as instrument, telescope, spectral range/resolution, and observational times for EOS observations in the revised manuscript (please see lines 757-761).

Lines 513-518: instruments? Resolutions (when not indicated)?

Reply: We have provided such information (e.g., instrument, spectral range/resolution, and observational times) for the three data sets (please see caption of Fig. S17, which was revised from the original Fig. S21 by following the suggestion from the other reviewer).

Lines 530-534: The authors justify some sources of the differences between the ESO and Cassini spectra. It is also noteworthy that the ESO spectrum of Saturn was acquired without rings.

Reply: We have added this important point (see lines 792-793).

Lines 536-537: Please specify the ESO instrument.

Reply: In a previous supplementary figure (Fig. S17), we have provided the details of EOS instrument (e.g., telescope, instrument, spectral range/resolution, and observational times). Then in the caption of this figure, we have stated that the EOS profile comes from Fig. S17.

Lines 710-711, 715-716: see comment about figures S16 and S17. Rephrase the sentence accordingly.

Reply: We rephrased these sentences (see lines 930-943). Additionally, please refer to our reply to your previous comment on Figs. S16 and S17 (now Figs. S13 and S14 in the revised manuscript).

Lines 750-752: specifications of the observations? See comment above regarding lines 346, 364, 371, 373, 394, 397.

Reply: As we replied to your previous comments, we have provided the image identification numbers for all ISS and VIMS images, which are currently used by the PDS, to facilitate searching and reproduction by other researchers (please see captions of Figs. S8, S9, S10, S11, S13, S14, S15, S22, and S23).

Line 785: lines 772-774 indicate that the authors assume that the Bond albedo does not significantly change between the two hemispheres. Rephrase the sentence which mentions <“probably” does not vary>.

Reply: It appears that the reviewer referred to line 795 instead of line 785, which contains the phrase “probably does not vary”. We have revised this sentence accordingly (please see line 1014).

REVIEWER COMMENTS

Reviewer #1 (Remarks to the Author):

The authors have done an excellent job in addressing the comments of my first review in great detail and exhaustive thoroughness. The paper has been fully restructured as I suggested and my main points of concern have been fully answered. Hence, I am delightedly to say that I think the revised paper satisfies all of Nature Communications requirements, and is wholly acceptable for publication.

Reviewer #2 (Remarks to the Author):

The authors have made significant improvements to the paper, resulting in enhanced readability and clarity. They have also successfully addressed all of my previous concerns. However, I do have a couple of minor comments on the updated version of the manuscript that should be addressed prior to publication.

In the updated version of the manuscript:

-Lines 336-345: mention or list other minor possible potential factors driving time variability, but not considered.

-Lines 757-761: please add the location of the telescope/instrument, i.e., La Silla.

-I still have a concern (related to wording) regarding the depiction of reflectance over time and the ability to discern any significant variability or trends. In the captions of Figures S13-S14 and lines 930-943, given the relatively large error bars, there is insufficient evidence to establish 'temporal variations.' Would it be appropriate to specify in the captions 'Saturn's full-disk reflectance across the recorded time by...'?

In reference to the text provided:

'Saturn's global and hemispheric reflectance, as well as its Bond albedo, exhibit relatively minor temporal fluctuations. Cassini ISS observations reveal that Saturn's full-disk reflectance varied approximately by 1.1-2.9% during the Cassini epoch (panel D of Figs. S13 and S14), a variance smaller than the ratio between the measurement uncertainty and the Bond albedo (~5%). Consequently, it appears that the temporal changes in Saturn's full-disk reflectance during the Cassini epoch lack statistical significance.'

Determining the significance of the observed trend/variability has enhanced the impact and credibility of the research finding. However, replacing 'exhibit relatively small temporal variations' with 'exhibit relatively minor temporal fluctuations' would be more precise. Additionally, instead of 'Therefore, the temporal variations of Saturn's full-disk reflectance during the Cassini epoch are not statistically significant,' a more suitable phrase could be: 'Consequently, the temporal changes in Saturn's full-disk reflectance during the Cassini epoch lack statistical significance.', for example.

REVIEWER COMMENTS

Reviewer #1 (Remarks to the Author):

The authors have done an excellent job in addressing the comments of my first review in great detail and exhaustive thoroughness. The paper has been fully restructured as I suggested and my main points of concern have been fully answered. Hence, I am delightedly to say that I think the revised paper satisfies all of Nature Communications requirements, and is wholly acceptable for publication.

Reviewer #2 (Remarks to the Author):

The authors have made significant improvements to the paper, resulting in enhanced readability and clarity. They have also successfully addressed all of my previous concerns. However, I do have a couple of minor comments on the updated version of the manuscript that should be addressed prior to publication.

In the updated version of the manuscript:

-Lines 336-345: mention or list other minor possible potential factors driving time variability, but not considered.

Reply: Yes, we have followed this suggesting by adding a sentence as below (please see lines 343-345 in the revised manuscript).

Other potential factors, such as small and mesoscale storms and waves, which may have relatively minor effects on the temporal variations of the radiant energy budget, are not considered in this study.

-Lines 757-761: please add the location of the telescope/instrument, i.e., La Silla.

Reply: Yes, we have added the location of European Southern Observatory (ESO) (La Silla, Chile) (please see lines 759-760 in the revised manuscript).

-I still have a concern (related to wording) regarding the depiction of reflectance over time and the ability to discern any significant variability or trends. In the captions of Figures S13-S14 and lines 930-943, given the relatively large error bars, there is insufficient evidence to establish 'temporal variations.' Would it be appropriate to specify in the captions 'Saturn's full-disk reflectance across the recorded time by...'?

Reply: Yes, we have modified the captions of the two figures in the Supplementary information by removing the phrase "temporal variations" (please see the lines 360-361 and 383-384 in the revised Supplementary Information).

In reference to the text provided:

'Saturn's global and hemispheric reflectance, as well as its Bond albedo, exhibit relatively minor temporal fluctuations. Cassini ISS observations reveal that Saturn's full-disk reflectance varied approximately by 1.1-2.9% during the Cassini epoch (panel D of

Figs. S13 and S14), a variance smaller than the ratio between the measurement uncertainty and the Bond albedo (~5%). Consequently, it appears that the temporal changes in Saturn's full-disk reflectance during the Cassini epoch lack statistical significance.'

Determining the significance of the observed trend/variability has enhanced the impact and credibility of the research finding. However, replacing 'exhibit relatively small temporal variations' with 'exhibit relatively minor temporal fluctuations' would be more precise. Additionally, instead of 'Therefore, the temporal variations of Saturn's full-disk reflectance during the Cassini epoch are not statistically significant,' a more suitable phrase could be: 'Consequently, the temporal changes in Saturn's full-disk reflectance during the Cassini epoch lack statistical significance.', for example.

Reply: We have followed the reviewer's constructive suggestions to do the corresponding replacements (please see lines 940 and lines 943-944 in the revised manuscript).

REVIEWERS' COMMENTS

Reviewer #2 (Remarks to the Author):

The authors have effectively addressed all the comments from my second review.

Therefore, I am pleased to state that I believe the revised version of the manuscript meets all the requirements of the journal, and is fully suitable for publication.